# SOCRATES LOSS FOR TRAINING AD-HOC CALIBRATED SELECTIVE CLASSIFIERS

## ABSTRACT

Model reliability is paramount for critical real-world applications. To enhance reliability, it is essential to quantify uncertainty in model predictions, as achieved through Confidence Calibration and Selective Classification. Confidence Calibration ensures prediction confidences accurately reflect the actual likelihood of correctness, while Selective Classification allows a model to abstain from making predictions when uncertain. Although related, existing methods address each aspect separately, or both through post-hoc methods. Only one method, Confidence-aware Contrastive Learning for Selective Classification (CCL-SC), combines both in an ad-hoc manner. Despite being a powerful calibrator, CCL-SC has some drawbacks, including the absence of an additional unknown class, the use of two different losses (detrimental for calibration), and its cumbersome implementation. In the pursuit of reliable models and motivated by the idea of creating an ad-hoc calibrated selective classifier with an unknown class, we first empirically analyze the Self-Adaptive Training (SAT) method, a leading method in ad-hoc selective classification. We identify that while SAT excels in selective classification, it falls short in confidence calibration, especially when training for a small number of epochs ($e.g., \leq 100$). To address this, we introduce an original method that uses an unknown class and a unique novel loss, *Socrates loss*, which serves as a classifier and a calibrator with a unified optimization goal. This method mitigates overfitting and ensures theoretically well-calibrated predictions across all epochs, addressing the drawbacks of both CCL-SC and SAT, without the need for post-hoc processing or additional data. We integrate our method into the SAT implementation and extend it to provide selective classification and confidence calibration metrics. We show empirically that our method matches or improves the selective classification error rate of SAT and CCL-SC, while producing well-calibrated models in an ad-hoc manner through the evaluation on 6 image benchmark datasets across two architectures, VGG-16 and ResNet-34.

## 1 INTRODUCTION

Reliability, the ability of a model to consistently operate in real-world environments (Tran et al., 2022), becomes particularly important in critical real-world scenarios, including but not limited to medical diagnosis (Gireesh & Gurupur, 2023), nuclear security (Ayodeji et al., 2022), and biosecurity (McEwen et al., 2021). A reliable model should not only achieve strong predictive performance but also excel in the representation of its own uncertainty. To quantify uncertainty, different methods measure distinct aspects of the predictive uncertainty stemming from reliability, such as Confidence Calibration and Selective Classification. Selective classification allows models to abstain from making predictions when uncertain, ensuring cautious decision-making in high-risk applications. On the other hand, confidence calibration ensures that predictive confidence accurately reflects the likelihood of correctness. Although both strategies aim to enhance reliability, they are typically approached and studied independently (Zhang et al., 2023).

In critical high-risk applications, where trust in predictions is essential, integrating confidence calibration with selective classification is crucial. Recent work has highlighted this need and proposed new post-hoc methods (Fisch et al., 2022; Galil et al., 2023; Moon et al., 2020), and, to the extent of our knowledge, only one ad-hoc method, Confidence-aware Contrastive Learning for Selective Classification (CCL-SC) (Wu et al., 2024). Despite the fact that CCL-SC is able to output cal-

ibrated models, it has several drawbacks. Firstly, following the work of Feng et al. (2023), the extra unknown class was not added. The use of an extra unknown class is related to adaptations aiming to address Open-Set Recognition (OSR) in deep neural networks (Bendale & Boult, 2016; Patrick Schlachter & Yang, 2019). In the search for reliable models, we argue that mechanisms for OSR should be integrated with calibration and selective classification to enhance model reliability and adaptability. A reliable model should not only adapt its predictions to new scenarios but also be flexible enough to handle different use cases. Incorporating an unknown class provides more flexibility, allowing the model to function as a selective classifier or a traditional classifier with or without an extra unknown class. Secondly, the CCL-SC method features two losses: Confidence-aware Supervised Contrastive (CSC) loss for calibration and cross-entropy (c.e.) loss for classification. We have identified that this method miscalibrates the model once it reaches a certain calibration point which is due to the c.e. effect (see Section 4.1.2). Therefore, having two losses can be detrimental if one of the losses is not specifically focused on calibration, highlighting the need for a unified loss with the same optimization goal. Thirdly, CCL-SC exhibits variable behavior in terms of Expected Calibration Error (ECE), depending on the architecture and dataset. The loss function across epochs also shows varied trends, including spikes, which could suggest training instability. Finally, the implementation of the CCL-SC method requires extensive modifications to the training code.

Motivated by these drawback and the idea of creating an ad-hoc calibrated selective classifier with a capability to estimate the probability for an unknown class, we first empirically analyzed the calibration capability of the Self-Adaptive Training (SAT) (Huang et al., 2020), the state-of-the-art for end-to-end selective classification with an extra unknown class. This analysis showed that SAT loss does not seem to be a calibration loss, as in the case of training for a smaller number of epochs (*e.g.*, $\leq 100$) or using hard-to-classify datasets like CIFAR-100 and Food-101.

We propose a method to train calibrated selective classifiers by introducing an extra unknown class and using a novel unified loss, *Socrates* loss. Socrates loss owes its name to the famous quote of the philosopher Socrates: *I know that I know nothing*; which reflects the power of the loss to train a model to be aware of its own uncertainty. This loss integrates classification and calibration into a single optimization problem, optimizing a single loss function, and does need several losses, switching to a different loss during training, or post-hoc processing. The loss uses its knowledge about when it does not know, and dynamically utilizes model predictions to guide training, by giving more attention to hard-to-classify instances.

We empirically evaluated our method to SAT and CCL-SC on the CIFAR-10, CIFAR-10C, CIFAR-100, CIFAR-100C, SVHN, and Food-101 datasets with VGG-16 and ResNet-34 architectures. In terms of calibration across epochs, our method outperforms SAT and is comparable to CCL-SC, while effectively addressing the previously discussed drawbacks of both methods. Moreover, our method achieves similar or lower Selective Classification error rates compared to CCL-SC and SAT, with notable improvements observed over SAT on the hard-to-classify CIFAR-100, CIFAR-100C and Food101 datasets. For instance, using the Food-101 dataset, the VGG-16 architecture, and $100\%$ of coverage, we achieve Selective Classification error rates ($\%$) of 26.93 for Socrates, 68.23 for SAT and 27.18 for CCL-SC.

To summarise, our contributions are as follows:

- An easy-to-implement ad-hoc method that uses an extra unknown class and a novel loss, Socrates loss, integrating classification and calibration into a unified optimization goal.

- A Python implementation to train Socrates, SAT and CCS-CL, and evaluate selective classification and confidence calibration. In addition, the code to reproduce the results of this paper is also provided.

- A theoretical analysis that proves Socrates loss a) forms a regularize upper bound in the Kullback-Leibler divergence, avoiding overconfident predictions and improving calibration. b) acts as a regularizer (of the network weights) when the model is sufficiently confident, avoiding miscalibration and overfitting.

- A comparative empirical analysis of Socrates, SAT, CCL-CS, in terms of calibration and selective classification performance, on 6 benchmark datasets across two network architectures.

## 2 RELATED WORK

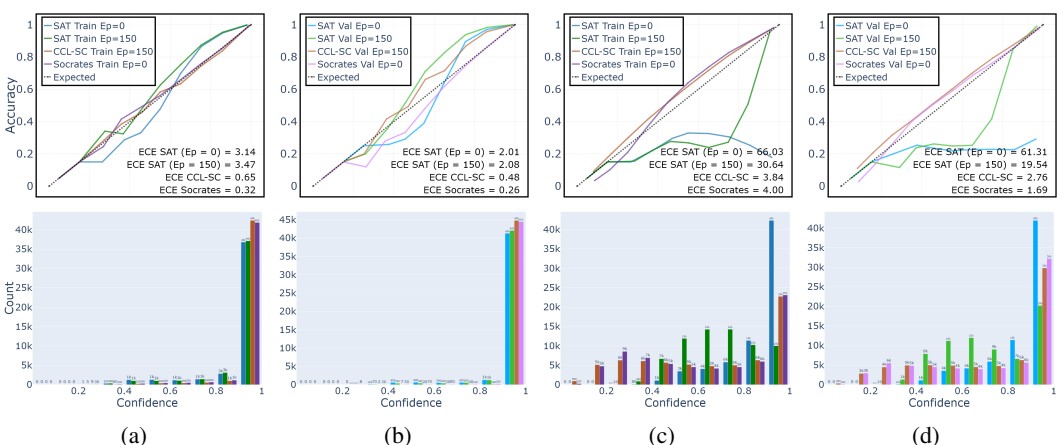

Figure 1: Reliability Diagrams of the last epoch displaying ECE values for CIFAR-10 (Figures 1a and 1b) and Food-101 (Figures 1c and 1d) datasets trained with the VGG-16 architecture using SAT, CCL-SC and Socrates methods.

In 2017, Guo et al. (2017) revealed that modern Neural Networks (NNs) are no longer well-calibrated and exhibit overconfidence. The degree of confidence calibration in NNs can be illustrated through visual representations and quantifiers (Section 3.2). To address the issue of miscalibration, the research community has focused on developing new post-hoc and ad-hoc methods. Post-hoc methods are applied to a trained model in a post-training process, such as Platt Scaling (Platt, 2000) and Temperature Scaling (Guo et al., 2017). Ad-hoc methods enhance both accuracy and calibration during training, creating end-to-end compact models by incorporating explicitly or implicitly a secondary optimization objective related to the predictive uncertainty of the model in the training objective (Liu et al., 2023). Although they remain underexplored, one way to achieve calibration is through the use of a specific loss function, such as Focal loss (Lin et al., 2020; Mukhoti et al., 2020). According to Zhang et al. (2023), despite the effectiveness of post-hoc methods, future models should integrate calibration into the training process. To that end, we focus our method on calibrating through a loss function.

Alternatively, one can train reliable models by considering the option to reject a prediction when the model is uncertain. Selective Classification (Geifman & El-Yaniv, 2017) can be addressed by post-hoc and ad-hoc methods. Post-hoc methods perform selective classification after training, such as LeCun et al. (1989) and Geifman & El-Yaniv (2017). Ad-hoc methods change the NNs training process and add extra heads or logits. Feng et al. (2023) divide these methods into learn to select, such as SelectiveNet (Geifman & El-Yaniv, 2019), and learn to abstain, such as Self-Adaptive Training (SAT) (Huang et al., 2020), methods. According to Feng et al. (2023), adding an extra head/logit is unnecessary for the ad-hoc Selective Classification problem, and the Softmax Response is the only selection mechanism required. We argue the opposite, that in the search for reliable models, the ability to handle unknown classes is valuable (Subsection 6.2).

Whereas some authors have emphasized the necessity of having calibrated selective classifiers (Zhang et al., 2023), the majority of published methods (Fisch et al., 2022; Galil et al., 2023; Moon et al., 2020) have focused on post-hoc integration, which presents several drawbacks: often requiring additional data, increasing the risk of bias, diffusing the optimization goal, and sometimes failing to fit when the calibration error is too complex. According to Zhang et al. (2023), a well-calibrated model could not be a good discriminator and vice versa. Although SAT was designed as a loss function to prevent overfitting, it has not been proven to be a promising ad-hoc calibration loss function across all epochs. Wu et al. (2024) presented CCL-SC, which is currently the state-of-the-art for ad-hoc calibrated selective classifiers. In contrast, Fisch et al. (2022) proposed a selective classifier that rejects instances based on calibration rather than potential misclassification, which represents a different goal from ours.

## 3 PROBLEM FORMULATION

We frame the problem as a multi-class classification task with $(c + 1)$ classes, where the last class represents the unknown class.

### 3.1 PROBLEM SETTING: SELECTIVE CLASSIFICATION

Selective Classification trades classifier coverage against accuracy. It is the ability of a model to reject instances when there is uncertainty. The rejected instances are potential out-of-distribution or lie in the tail of the data distribution; making predictions only on samples with confidence.

Let $\mathcal{X}$ be the feature space, $\mathcal{Y}$ be the label space, and $P(\mathcal{X}, \mathcal{Y})$ be the data distribution over $\mathcal{X} \times \mathcal{Y}$. A selective model is a pair $(f, g)$, where the prediction function is $f : \mathcal{X} \to \mathcal{Y}$, in our case a classifier, and the selection function is $g : \mathcal{X} \to \{0, 1\}$. Then, $f(x)$ makes a prediction when $g(x) = 1$, and abstains from making a prediction when $g(x) = 0$.

The performance of a selective classifier can be evaluated in terms of cost-sensitive learning (Cortes et al., 2016) where the rejection cost needs to be specified, or from a Risk-Coverage perspective (El-Yaniv & Wiener, 2010). Since specifying costs can be challenging (Geifman & El-Yaniv, 2017), we evaluate our method using the Risk-Coverage perspective. Coverage is defined as the probability mass of the nonrejected region of $\mathcal{X}$, $\phi(g) = \mathbb{E}[g(X)]$. In practice, a soft selection function $\tilde{g} : \mathcal{X} \to \mathbb{R}$ is often used, constraining the coverage with a threshold $\tau \in \mathbb{R}$. Then $g$ is defined as $g(x) := \mathbf{1}\{\tilde{g}(x) \geq \tau\}$. Given a loss function, the selective risk, which corresponds to the selective error when the loss is 0/1, with respect to $P$ can be defined as $R(f, g) = \mathbb{E}[\mathcal{L}(f(X), Y) \mid g(X) = 1] = \frac{\mathbb{E}[\mathcal{L}(f(X), Y) \mid g(X)]}{\phi(g)}$. This shows a dependency between risk and coverage; rejecting samples results in lower selective risk and lower coverage. Therefore, from a Risk-Coverage perspective, the minimization problem given a target coverage is: $\min R(f, g)$ s.t. $\phi \geq c_{\text{target}}$.

We follow the Selective Classification problem for ad-hoc methods to train end-to-end selective classifiers proposed by SAT (Huang et al., 2020) and DeepGamblers (Liu et al., 2019), where the selection function $g(\cdot)$ is replaced by $f(\cdot)_c$ where $c$ is the number of classes. In our proposed method, similar to SAT and DeepGamblers, an additional unknown class $(c + 1)$ represents abstention.

### 3.2 PROBLEM SETTING: CONFIDENCE CALIBRATION

Confidence calibration is the process of aligning predictive confidence with the actual likelihood of correctness, i.e. accuracy in the multiclass case. One method to reach ad-hoc calibration is through a loss function as with Focal loss (Lin et al., 2020). The confidence calibration level of a NN can be represented through visualizations and quantifiers.

A popular method for visualising confidence calibration is the Reliability Diagrams (Niculescu-Mizil & Caruana, 2005), which plot the expected sample accuracy as a function of confidence. Confidences can be grouped in different forms (Filho et al., 2023; Guo et al., 2017; Nguyen & O'Connor, 2015) to estimate expected accuracy from finite samples. In this work, we adopt the approach of Guo et al. (2017), grouping confidences into $M$ interval bins of size $1/M$, increasing the probability of having multiple samples per estimation range. Let $B_m$ be the test set of indices of samples whose confidence falls into the $m$-th bin, $I_m = (\frac{m-1}{M}, \frac{m}{M}]$. The confidence of bin $B_m$ is estimated as $conf(B_m) = \frac{1}{|B_m|} \sum_{i \in B_m} \hat{p}_i$; where $\hat{p}_i$ is the confidence for sample $i$. The average accuracy is estimated as $acc(B_m) = \frac{1}{|B_m|} \sum_{i \in B_m} \mathbf{1}(\hat{y}_i = y_i)$; where $\hat{y}_i$ is the predicted class label and $y_i$ is the true class label for sample $i$.

To measure calibration the most popular metrics are the Expected Calibration Error (ECE) and the Maximum Calibration Error (MCE) (Naeini et al., 2015). ECE is the weighted average of the difference between accuracy and confidence in each bin: $ECE = \sum_{m=1}^{M} \frac{|B_m|}{n} |acc(B_m) - conf(B_m)|$. MCE is the worst-case deviation and is valuable for high-risk frameworks: $MCE = \max_{m \in \{1, ..., M\}} |acc(B_m) - conf(B_m)|$. It is common to use Brier Score as in Fisch et al. (2022) but, as it is an aggregate measure (Hernández-Orallo et al., 2012), it is inadequate for analyzing calibration in isolation.

An example of Reliability Diagrams along with the ECE values is presented in Figure 1.

# 4 OUR METHOD: CALIBRATED SELECTIVE CLASSIFICATION WITH AN UNKNOWN CLASS

We propose a versatile method that can be used as a selective classifier or as a standard classifier with or without an unknown class; well-calibrated in all cases. Inspired by the calibration principles of Focal loss and influenced by the selective classification power of SAT, we introduce a method to train calibrated selective classifiers by integrating an additional unknown class, referred to as $idk$, and using an easy-to-implement novel loss called Socrates loss, which maintains a unified optimization objective of classification and calibration.

Therefore, a classifier $f(\cdot)_c$ is optimized by minimizing Socrates loss, which is defined as:

$$\mathcal{L}(f) = -\frac{1}{n} \sum_{i=1}^{n} (1 - \hat{p}_{i,y_i})^\gamma [t_{i,y_i} \log \hat{p}_{i,y_i} + \alpha_{dynamic}(1 - t_{i,y_i}) \log \hat{p}_{i,idk}]. \tag{1}$$

where $\hat{p}_{i,y_i}$ is the prediction associated with the ground truth class and $\hat{p}_{i,idk}$ is the prediction associated with the *idk* class, $n$ the number of instances, $\gamma$ a modularity factor controlling the downweighting of easy examples (higher factor gives more weight to hard-examples), and $\alpha_{\text{dynamic}}$ is a regularizer which controls attention to the unknown knowledge.

Initially, during the first selected $E_s$ training epochs, the target is the ground truth label, $t_i \leftarrow y_i$. After $E_s$, the target is updated in each epoch as $t_i \leftarrow \alpha_{\text{momentum}} \times t_i + (1 - \alpha_{\text{momentum}}) \times \hat{p}_{i,y_i}$ s.t. $\alpha_{\text{momentum}} \in (0, 1]$. This dynamic behaviour balances the importance of current predictions associated with the ground truth class and the *idk* class, reducing prediction instability. Our main proposal uses $E_s = 0$, thereby creating an end-to-end loss.

The logic behind the loss can be described as follows. If a sample was previously predicted with high confidence, the first part of the equation has more influence, resembling Focal loss and giving a bigger penalty towards hard-to-classify samples. This method helps to avoid overfitting and calibrate the model when the uncertainty is low. Conversely, if the sample seems uncertain (i.e., low previous confidence), the second part of the equation assumes greater importance acting as a selection function in the selective classifier. This part is influenced by an $\alpha_{\text{dynamic}} \leftarrow (\max_{y_i \neq y_{gt}} \hat{p}_{i,y_i}) - \hat{p}_{i,y_{idk}}$, which adjusts attention based on the awareness of the classifier of its own uncertainty, of its own lack of knowledge. If the classifier recognizes its own uncertainty, i.e., the idk class predicted probability surpasses other class probabilities (without the ground truth class probability), only the first part is considered; as the model knows it does not know. Otherwise, if the classifier is not aware of its lack of knowledge, the selection function gains relevance weighted by the focal component. This method increases penalties for hard-to-classify instances and for instances where the classifier does not have certainty that it does not know.

The pseudocode of the method can be found in Appendix A and a mathematical example in B.

## 4.1 THEORETICAL ANALYSIS

### 4.1.1 SOCRATES LOSS FORMS A REGULARIZED UPPER BOUND IN THE KULLBACK-LEIBLER DIVERGENCE

It is well-known that c.e. loss minimizes (provides an upper bound for) the Kullback-Leibler (KL) divergence between the predicted and the target distributions over classes, i.e., $\mathcal{L}_{\text{c.e.}}(f) \geq D_{\text{KL}}(q||\hat{p})$. KL divergence quantifies the information difference between two distributions. In our case, Socrates loss minimizes KL divergence while regularizing by increasing the entropy of the predicted distribution and leveraging the predictions associated with the unknown class. The regularization parameters are $\gamma, \alpha_{\text{dynamic}}$, and $\triangle_{reg}$; where $\triangle_{reg} = (1 - t_y)[\gamma \hat{p}_y \log \hat{p}_{idk} - \log \hat{p}_{idk}]$. Therefore:

$$\mathcal{L}(f) \geq D_{\text{KL}}(q||\hat{p}) - \gamma \mathbb{H}[\hat{p}] + \alpha_{\text{dynamic}} \triangle_{reg}. \tag{2}$$

This regularised entropy increase, along with the regularization applied through the prediction associated with the unknown class, prevents the model from becoming overconfident. Then, substituting

the c.e. loss with Socrates loss incorporates a maximum-entropy regulariser (Pereyra et al., 2017) to the KL minimization. As demonstrated by Lin et al. (2020), higher entropy can prevent overconfident predictions, improving model calibration. Therefore, Socrates Loss forms a regularize upper bound in the KL divergence, avoiding overconfident predictions and improving calibration. The proof can be found in Appendix C.1.

### 4.1.2 SOCRATES LOSS REGULARIZES THE WEIGHTS OF THE NETWORK

Guo et al. (2017) and Lin et al. (2020) proved there is a relationship between miscalibration and overfitting (but not the opposite). This occurs when the loss function attempts to further reduce its value even after perfect high confidence has been achieved. Lin et al. (2020) demonstrated that for misclassified samples using c.e. loss the network progressively grows more confident in its incorrect predictions. Socrates loss acts as a regularizer with an increased penalty highly associated with the unknown class when the model begins to overfit. Furthermore, the norms of the weights are higher at the beginning of the training compared to those trained with c.e. It is when the model starts being miscalibrated that there is a change in the ordering of the weight norms, due to a big increase in the weight norm of the models with c.e. This behaviour shows that Socrates loss acts as a regularizer when the model is sufficiently confident, avoiding miscalibration and overfitting.

Therefore, let $\mathcal{L}_{\text{c.e.}}(f)$ be c.e. loss, and $\mathcal{L}(f)$ be Socrates loss. The gradients of the neural network trained with $\mathcal{L}(f)$ are smaller than the ones trained with $\mathcal{L}_{\text{c.e.}}(f)$ when a perfect confidence is reached and the model could start overfitting and then become miscalibrated, i.e.,

$$||\frac{\partial \mathcal{L}(f)}{\partial w}|| \leq ||\frac{\partial \mathcal{L}_{\text{c.e.}}(f)}{\partial w}||. \tag{3}$$

The proof can be found in Appendix C.2.

## 5 EXPERIMENT SETTINGS

For the upcoming experiments, we initially evaluated SAT against Focal loss, a calibration loss function. Afterwards, we evaluated the Socrates method, comparing it to the SAT and CCL-SC methods. To this end, we extended the publicly available SAT implementation to create a framework for training and evaluating calibrated selective classifiers.[1]

Table 1: Specifications of datasets employed in the experimental phase.

| Dataset | Image Size | Classes | Train | Test | Specifications |
|---------|-----------|---------|-------|------|----------------|
| CIFAR-10 | 32x32x3 | 10 | 50000 | 10000 | Easy-to-classify |
| CIFAR-10C | 32x32x3 | 10 | | 50000 | 10000 using 5 levels of corruption |
| CIFAR-100 | 32x32x3 | 100 | 50000 | 10000 | Hard-to-classify dataset |
| CIFAR-100C | 32x32x3 | 100 | | 50000 | 10000 using 5 levels of corruption |
| SVHN | 32x32x3 | 10 | 73257 | 26032 | Easy-to-classify real-world dataset |
| Food-101 | 224x224x3 | 101 | 75750 | 25250 | Hard-to-classify real-world dataset |

We used a VGG-16 and a ResNet-34 architecture for the datasets specified in Table 1. Each configuration was trained with five different seeds. Additional hyper-parameters details are in Appendix D.

SAT, CCL-SC and Socrates methods can be initialized in the first epochs with another loss (e.g., c.e. or Focal loss), and then switched to the main loss. For Selective Classification, the SAT authors instantiated the number of first epochs at 0, and for CCL-SC at 150. Therefore, we conducted two experiments: *first-epochs* ($E_s = 150$) for Socrates and SAT methods, and *end-to-end* ($E_s = 0$) for Socrates with Focal and Socrates losses, SAT with c.e. and SAT losses, and CCL-SC with c.e. and CSC losses. The goal is to determine whether a method with a unified loss, i.e., *first-epochs* case with Socrates method using only the Socrates loss, can achieve or surpass similar selective classification results while addressing the calibration issues of SAT and CCL-SC.

---

[1]The code is publicly available at `https://anonymous.4open.science/r/Socrates`

# 6 RESULTS

## 6.1 IS SELF-ADAPTIVE TRAINING LOSS A CALIBRATION LOSS?

To our understanding, the SAT method achieves the highest performance in the Selective Classification problem with an unknown class. The first question to address is: *Is Self-Adaptive Training a calibrated loss?* Since SAT adds an extra unknown class and modifies the loss function to alleviate overfitting, it is reasonable to consider SAT loss as a potential calibration loss similar to Focal loss. However, the role of SAT as a calibrator has not been explored in the literature. For this initial empirical analysis, we set aside the Selective Classification problem and focus solely on the confidence calibration problem.

A detailed analysis with graphs is presented in Appendix F. To sum up, first, we observed that the accuracy and loss across epochs curves for Focal Loss exhibited similar trends, with minor overfitting noted in the Food-101 dataset during the initial epochs. In contrast, the SAT loss demonstrated different trends and did not consistently prevent overfitting. Regarding calibration metrics, the ECE across epochs showed a consistent downward trend for Focal Loss, except for VGG-16 when applied to the CIFAR-100 and Food-101 datasets, where an increase was observed in the initial epochs but remained within an acceptable ECE range. For SAT loss and VGG-16, a rise was observed in ECE after the 150 epochs in the first epochs decreasing after convergence, while the rise for ResNet-34 was less discernible. In the SAT end-to-end case, ECE values were notably high during the initial epochs for both architectures. The MCE across epochs displayed similar trends for Focal Loss, but distinct trends were observed for SAT. Importantly, SAT does not appear to be an effective calibration loss and may be detrimental when the goal is to train for a small number of epochs ($\leq 100$), as it outputs calibrated confidences only after a considerable amount of training epochs. Additionally, we noted that the average confidence values of the *idk* class seem to be directly related to calibration, reflecting similar trends as the ECE across epochs. This raised the question *Might the additional idk class method be beneficial or detrimental in terms of calibration?* This observed behavior was the main source of inspiration for incorporating predictions associated with the *idk* class into the novel Socrates loss to calibrate during training.

Based on the empirical analysis the following claim can be made: Unlike Focal loss, which produces very well-calibrated models and follows similar trends across all the datasets and architectures, SAT loss exhibits certain tendencies that ultimately lead to the conclusion that it is not a loss that allows learning calibrated models in all the epochs and scenarios, especially when aiming to train for a small number of epochs or when dealing with complex datasets such as Food-101. Additionally, when the loss is used *end-to-end*, the miscalibration in the first epochs is excessively large, and in some cases (CIFAR-100 and Food-101 with VGG-16) it remains significantly large until the end of training. When the loss is applied with *first-epochs* case, miscalibration begins to emerge. Therefore, we can claim that SAT loss seems not to be a calibration loss.

## 6.2 SOCRATES LOSS AS A CALIBRATOR

Before addressing the topic of Selective Classification, a similar question asked in Section 6.1 needs to be considered: *Is the novel Socrates loss a calibrated loss?* To investigate this, the same methodology of Section 6.1 is followed. Since SAT (*end-to-end* and *first-epochs*) has been empirically shown not to be a suitable calibration loss, our novel method (*end-to-end*) is compared with the CCL-SC method (*first-epochs* case as CCL-SC has two losses).

Due to space reasons, the curves for the SVHN and CIFAR-100 datasets, along with those for CIFAR-10 and Food-101, are presented in Appendix G.

**Overfitting:** In contrast to the SAT method, both Socrates and CCL-SC effectively mitigate the overfitting issue, improving generalization across all three datasets and both architectures. This preliminary empirical analysis suggests that Socrates loss may be a prominent calibration loss. In fact, the accuracies achieved with this novel loss outperform those obtained with the SAT loss, showcasing a substantial improvement. When comparing Socrates with CCL-SC, the accuracies are similar in most scenarios, except for the SVHN dataset with the VGG-16 architecture. Here, Socrates achieves accuracies close to $100\%$, while CCL-SC reaches approximately $80\%$. Notably, Socrates consistently exhibits a downward trend in output losses across all scenarios, whereas CCL-

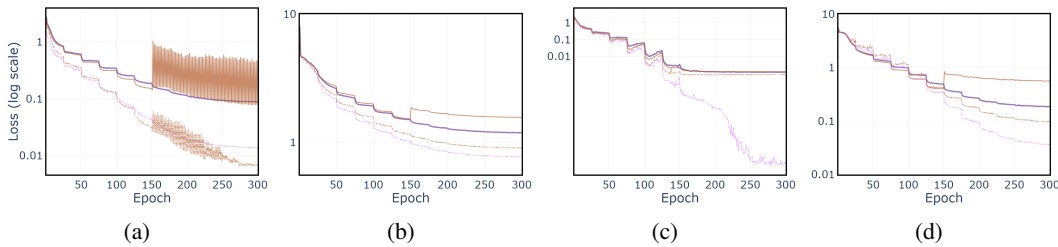

Figure 2: Loss curves of models trained on CIFAR-10 (a and c) and Food-101 (b and d) datasets using using Socrates and CCL-SC methods with VGG-16 (a and b) and ResNet-34 (c and d) architectures.

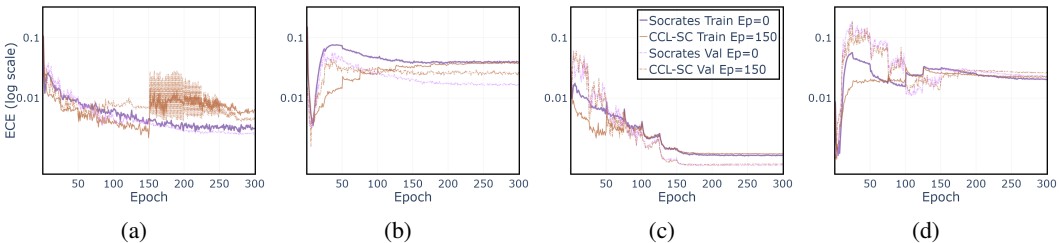

Figure 3: Evolution of the Expected Calibration Error (ECE) across epochs for models trained on CIFAR-10 (a and c) and Food-101 (b and d) datasets using Socrates and CCL-SC methods with VGG-16 (a and b) and ResNet-34 (c and d) architectures.

SC shows varied trends, including significant spikes and upward and downward trends depending on the architecture and dataset. Since both trainings use the same seeds and hyperparameters, except for the loss function, the observed spikes in CCL-SC suggest potential instability in training. The loss curves for CIFAR-10 and Food-101 are presented in Figure 2.

**Calibration Metrics:** The reliability diagrams with the ECE of the last epoch (Figure 1) do not provide enough information to draw calibration conclusions, instead, the ECE and MCE values along the epochs produce noticeable insights. In the first place, the ECE value along epochs is carried out. The values for CIFAR-10 and Food-101 are visualized in Figure 3.

Whereas SAT performs differently for each architecture and case, Socrates exhibits consistent trends across both architectures, showing a significant drop in ECE values after the initial epochs. Although Socrates shows an initial fluctuation in ECE values across epochs (which varies depending on the difficulty of the dataset), the ECE values across all epochs are consistently low, within a range below 10%, suggesting that the model is well-calibrated. Socrates achieves better ECE values than SAT across all epochs, datasets, and architectures.

When comparing Socrates with CCL-SC, both methods achieve similar ECE values. However, CCL-SC exhibits certain drawbacks. First, depending on the dataset and architecture, CCL-SC features spikes as equal to the loss across epochs. Second, while Socrates consistently shows an initial fluctuation followed by a decrease in ECE values, CCL-SC begins to miscalibrate the model once it reaches a lower ECE point. Although the ECE values remain within a small range, the upward trend indicates that CCL-SC could lead to miscalibrated models. Given that CCL-SC employs two losses (CCL loss for calibration and cross-entropy loss for classification), we argue that the calibration detriment could be attributed to the cross-entropy loss, which may miscalibrate the model by attempting to further reduce the loss after achieving the ideal confidence, thereby increasing the weight norm (Section 4.1.2). Therefore, having multiple losses could be detrimental if one of the losses is not specifically focused on calibration. This raises the question of why use several losses if a unified loss function can achieve the same goals, such as Socrates loss.

**Idk class:** Socrates reaffirms the claim made in Section 6.1: there is a link between the ECE values and the average of the confidences associated with the idk class. Moreover, addressing the question

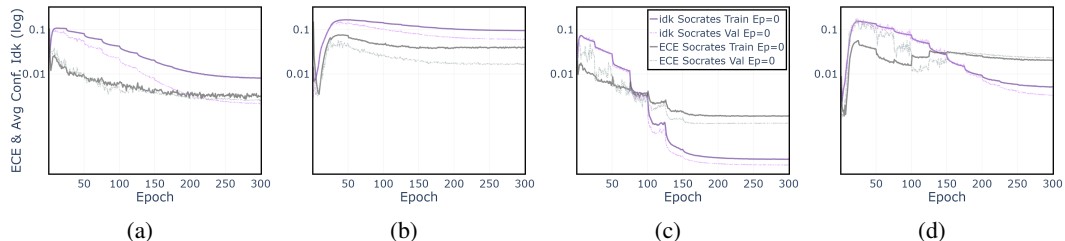

(a)        (b)        (c)        (d)

Figure 4: Curves depicting the average values of the idk class confidences across the epochs and ECE across epochs of models trained on CIFAR-10 (a and c) and Food-101 (b and d) datasets using Socrates method with VGG-16 (a and b) and ResNet-34 (c and d) architectures.

*Might the extra idk class approach be beneficial for calibration, or could it be detrimental?* we argue that incorporating the extra *idk* class and introducing $\alpha_{\text{dynamic}}$ in the loss function offers a distinct advantage for calibrated selective classifiers. This mechanism, which is in Socrates loss, helps the model adjust penalization based on the confidence levels of its predictions. The curves showing the average values of the *idk* class confidences across epochs and the ECE across epochs for CIFAR-10 and Food-101 are presented in Figure 4.

**Socrates loss is a suitable loss to output calibrated models:** These findings underscore the effectiveness of Socrates as an end-to-end calibration method for training models, particularly when only a small number of epochs (in contrast to SAT, which is not suitable) are required to train trustworthy outputs in terms of confidence calibration and when an unknown class is considered. The ability of Socrates to function without having multiple losses allows for a unified optimization goal, simultaneously addressing both classification and calibration in an ad-hoc manner.

### 6.2.1 SOCRATES LOSS AS A SELECTIVE CLASSIFIER

This paper focuses on calibrating selective classifiers, aiming to produce ad-hoc calibrated selective classifiers suitable for deployment in real-world critical environments. While improving Selective Classification error rates was not the primary goal of our study, which was more focused on enhancing calibration, Socrates demonstrates improvements over SAT on challenging datasets such as Food-101. In comparison with CCL-SC, both methods achieve comparable performance. There are instances where Socrates outperforms, as in SVHN with VGG-16, where Socrates achieves an error rate close to 3% compared to around 18% for CCL-SC and SAT for the *first-epochs* case. The most relevant results are presented in Table 2, and for space reasons in Appendix E. The risk-coverage curves provide a clear demonstration of the strength of the Socrates method compared to CCL-SC. These curves reveal that Socrates consistently achieves similar or better values than CCL-SC. The detailed Risk-Coverage curves can be found in Figure 5 and Appendix H, where a notable improvement is observed, particularly for the CIFAR-10 and SVHN datasets.

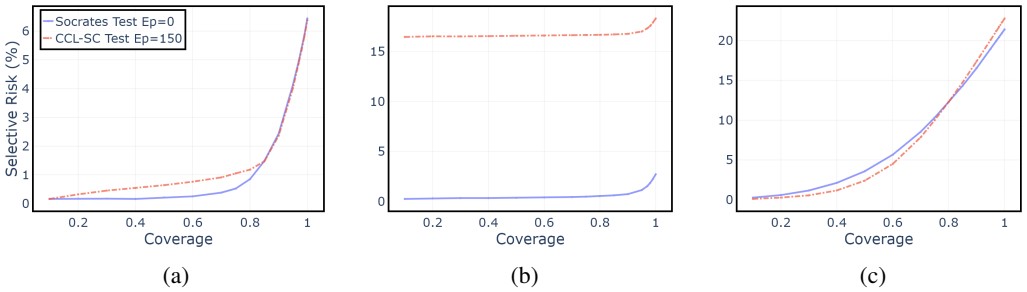

(a)         (b)         (c)

Figure 5: Risk-Coverage curves of models trained on CIFAR-10 (a), SVHN (b) and Food-101 (c) datasets using Socrates (*end-to-end* case) and CCL-SC (*first-epochs* case) methods with VGG-16 (a and b) and ResNet-34 (c) architectures.

Table 2: Selective Classification error rates % on CIFAR-10, SVHN and Food-101 datasets for various coverage rates %, reported with mean and standard deviation. Underline indicate the overall best performance, while bold highlight the best performance in each case.

| Dataset | Coverage | *end-to-end* case | | *first-epochs* case | | |
| --- | --- | --- | --- | --- | --- | --- |
| | | Socrates (ours) | SAT | Socrates + Focal | CCL-SC + c.e | SAT + c.e |
| | 100 | **6.44 ± 0.18** | 7.08 ± 1.07 | 6.67 ± 0.19 | *6.38 ± 0.14* | 6.87 ± 1.08 |
| | 95 | **4.14 ± 0.12** | 4.78 ± 0.98 | 4.45 ± 0.14 | *4.02 ± 0.14* | 4.58 ± 1.12 |
| | 90 | **2.43 ± 0.09** | 3.01 ± 0.88 | 2.76 ± 0.13 | *2.36 ± 0.13* | 2.92 ± 1.01 |
| CIFAR-10 | 85 | **1.48 ± 0.11** | 1.82 ± 0.65 | 1.64 ± 0.20 | *1.47 ± 0.16* | 1.75 ± 0.74 |
| VGG-16 | 80 | **0.85 ± 0.03** | 1.12 ± 0.51 | *1.05 ± 0.11* | 1.18 ± 0.25 | 1.05 ± 0.46 |
| | 75 | 0.52 ± 0.03 | 0.67 ± 0.32 | *0.68 ± 0.07* | 1.05 ± 0.19 | **0.61 ± 0.27** |
| | 70 | 0.38 ± 0.04 | 0.43 ± 0.24 | *0.51 ± 0.05* | 0.91 ± 0.11 | **0.42 ± 0.20** |
| | 100 | 2.72 ± 0.07 | **2.65 ± 0.04** | *2.80 ± 0.03* | 18.29 ± 34.73 | 18.22 ± 34.77 |
| | 95 | 1.15 ± 0.04 | **1.04 ± 0.02** | *1.20 ± 0.08* | 16.99 ± 35.46 | 16.89 ± 35.51 |
| | 90 | 0.74 ± 0.05 | **0.61 ± 0.05** | *0.80 ± 0.05* | 16.76 ± 35.58 | 16.57 ± 35.69 |
| SVHN | 85 | 0.62 ± 0.02 | **0.45 ± 0.04** | *0.62 ± 0.05* | 16.70 ± 35.62 | 16.44 ± 35.76 |
| VGG-16 | 80 | 0.55 ± 0.03 | **0.38 ± 0.02** | *0.54 ± 0.05* | 16.66 ± 35.64 | 16.39 ± 35.79 |
| | 75 | 0.49 ± 0.05 | **0.33 ± 0.02** | *0.51 ± 0.03* | 16.64 ± 35.65 | 16.35 ± 35.81 |
| | 70 | 0.45 ± 0.04 | **0.30 ± 0.01** | *0.48 ± 0.02* | 16.62 ± 35.66 | 16.33 ± 35.82 |
| | 100 | **21.40 ± 0.79** | 100 ± 0.0 | 32.33 ± 22.32 | *22.77 ± 0.90* | **22.08 ± 0.75** |
| | 95 | **18.95 ± 0.80** | 100 ± 0.0 | 30.20 ± 23.10 | *20.09 ± 0.92* | **20.02 ± 0.74** |
| | 90 | **16.54 ± 0.75** | 100 ± 0.0 | 28.23 ± 23.92 | *17.39 ± 0.91* | 17.97 ± 0.74 |
| Food-101 | 85 | **14.32 ± 0.74** | 100 ± 0.0 | 26.37 ± 23.92 | *14.75 ± 0.92* | 15.99 ± 0.72 |
| ResNet-34 | 80 | **12.30 ± 0.78** | 100 ± 0.0 | 24.60 ± 25.11 | *12.30 ± 0.94* | 14.08 ± 0.67 |
| | 75 | **10.32 ± 0.68** | 100 ± 0.0 | 22.94 ± 25.57 | *10.00 ± 0.81* | 12.20 ± 0.64 |
| | 70 | **8.54 ± 0.62** | 100 ± 0.0 | 21.49 ± 25.97 | *7.85 ± 0.70* | 10.37 ± 0.60 |

# 7 CONCLUSIONS AND LIMITATIONS

In this paper, we first empirically investigated the calibration capacity of SAT loss as a calibration mechanism, finding that it does not produce well-calibrated models. This deficiency is particularly detrimental for models that require only a small number of epochs or when working with hard-to-classify datasets. Additionally, we found that SAT does not consistently mitigate overfitting across all cases. Through this empirical study, we identified a relationship between the extra unknown class and calibration, which inspired the development of our proposed loss function. To address the need for ad-hoc easy-to-implement calibrated selective classifiers with an unknown class, we proposed a new method that incorporates an extra unknown class and introduces a novel loss, *Socrates*, with a unified optimization goal (classification and calibration). We theoretically and empirically analyzed this loss, demonstrating that it is an optimal calibration method without the previously enumerated drawbacks of SAT and CCL-SC. This new loss not only ensures strong calibration throughout all training epochs (making it suitable for models trained with fewer epochs), but also produces selective classifiers that achieve similar Selective Classification error rates to SAT and CCL-SC, while notably outperforming SAT on hard-to-classify datasets such as CIFAR-100 and Food-101 and CCL-SC on datasets such as SVHN for VGG-16.

We encourage the research community to further evaluate the Socrates method across a broader spectrum of architectures and datasets. It is notable that this method has not been compared to post-hoc methods. We argue that the strength of this end-to-end method comes from producing compact models that do not require post-processing and additional data. Leveraging all available data during training can be particularly advantageous when data is limited. Future research should incorporate and evaluate additional reliability aspects to develop a more comprehensive reliability framework (e.g., distribution shifts, noise, out-of-distribution, etc.). The lack of metrics that integrate reliability concepts is a pressing need. For example, a model may often be well-calibrated but exhibit low accuracy. Additionally, there is a need for metrics that summarize calibration performance across epochs. For example, a new ECE-epochs metric could indicate whether calibration has improved or deteriorated at any given point.

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

APPENDIX: TRAINING RELIABLE MODELS:
HAVING THE CONFIDENCE TO SAY "I DON'T KNOW"

## A  PSEUDOCODE FOR THE SOCRATES METHOD

---

**Algorithm 1** Training with Socrates loss

---

**Require:** Data $\{(x_i, y_i)\}_{i=1}^n$, initial targets $\{t_i\}_{i=1}^n = \{y_i\}_{i=1}^n$, initial model $f$, batch size $bs$, and hyper-parameters: momentum term $\alpha_{\text{momentum}}$, modularity factor $\gamma$, and initial epochs $E_s$.

1: **repeat**
2:    **for** e = 1 **to** maximum_epochs **do**
3:        **for** each mini-batch data $\{(x_i, y_i)\}_{bs}$ in the current epoch e **do**
4:            **for** i = 1 **to** bs **(in parallel) do**
5:                $\hat{p}_i = softmax(f(x_i))$
6:                $\alpha_{\text{dynamic}} \leftarrow \big(\max_{y_i \neq y_{gt}} \hat{p}_{i,y_i}\big) - \hat{p}_{i,y_{idk}}$
7:                **if** $e \geq E_s$ **then**
8:                    $t_i \leftarrow \alpha_{\text{momentum}} \times t_i + (1 - \alpha_{\text{momentum}}) \times \hat{p}_{i,y_i}$
9:                **end if**
10:               $\mathcal{L}(f) = -\frac{1}{n} \sum_{i=1}^n (1 - \hat{p}_{i,y_i})^\gamma [t_{i,y_i} \log \hat{p}_{i,y_i} + \alpha_{dynamic}(1 - t_{i,y_i}) \log \hat{p}_{i,idk}]$
11:               Update the weights of $f$ using an optimizer based on $\mathcal{L}(f)$
12:           **end for**
13:       **end for**
14:   **end for**
15: **until** end of training

---

Although our method can be used with other losses due to the flexibility of the initial epochs variable, our primary goal is to design an end-to-end loss. Therefore, we set $E_s = 0$ in our main results.

## B  MATHEMATICAL EXAMPLE OF THE SOCRATES METHOD

To illustrate how Socrates loss operates, consider a selective classifier with $E_s = 0$, $\gamma = 2$, and $\alpha_{\text{momentum}} = 0.9$, which can output one of three classes: predator, non-predator, or idk. We will examine the following three scenarios:

1. An image of a cat with a ground truth label of predator. The classifier outputs the confidences $[0.9, 0.05, 0.05]$ at epoch 30, and $[0.9, 0.02, 0.08]$ at epoch 31. Since the previous prediction (epoch 30) had high confidence, the $t_i = 0.9$. For epoch 31, as $\max_{y_i \neq y_{gt}} \hat{p}_{i,y_i}$ corresponds to the idk class, $\alpha_{\text{dynamic}} = 0$. Therefore, only the first part of the loss function is relevant, giving more penalty to hard-to-classify instances. The loss at epoch 31 is $\mathcal{L} = 0.0009$.

2. An image of a fake pink cat with a ground truth label of predator. The classifier outputs the confidences $[0.5, 0.25, 0.25]$ at epoch 30, and $[0.5, 0.3, 0.2]$ at epoch 31. Since the previous prediction (epoch 30) lacked high confidence, both parts of the equation are relevant, $t_i = 0.5$. In this case, $\max_{y_i \neq y_{gt}} \hat{p}_{i,y_i}$ is the non-predator class, then $\alpha_{\text{dynamic}} = 0.1$; the model is unaware of its lack of knowledge. The loss at epoch 31 is $\mathcal{L} = 0.11$.

3. An image of a fake pink cat with a ground truth label of predator. The classifier outputs the confidences $[0.5, 0.25, 0.25]$ at epoch 30, and $[0.5, 0.2, 0.3]$ at epoch 31. As the previous prediction (epoch 30) lacked high confidence, both parts of the equation take relevance, $t_i = 0.5$. In this case, $\max_{y_i \neq y_{gt}} \hat{p}_{i,y_i}$ is the idk class, then $\alpha_{\text{dynamic}} = 0$; the model is aware of its lack of knowledge. The loss at epoch 31 is $\mathcal{L} = 0.088$.

## C  THEORETICAL PROOFS

### C.1  SOCRATES LOSS FORMS A REGULARIZED UPPER BOUND IN THE KULLBACK-LEIBLER DIVERGENCE

**Theorem:**  Socrates loss minimizes (creates an upper bound for) the Kullback-Leibler (KL) divergence while regularizing by increasing the entropy of the predicted distribution and leveraging the predictions associated with the unknown class. The regularization parameters are $\gamma, \alpha_{\text{dynamic}}$, and $\triangle_{reg}$; where $\triangle_{reg} = (1 - t_y)[\gamma \hat{p}_y \log \hat{p}_{idk} - \log \hat{p}_{idk}]$. Therefore:

$$\mathcal{L}(f) \geq D_{\text{KL}}(q\|\hat{p}) - \gamma \mathbb{H}[\hat{p}] + \alpha_{\text{dynamic}} \triangle_{reg}; \tag{4}$$

**Proof:**  Let the KL divergence be the divergence between the ground truth distribution $q$ and the predicted distribution $\hat{p}$, and $\mathbb{H}[q]$ be the entropy of the ground truth distribution defined as $\mathbb{H}[q] = -\sum_j q_j \log(q_j)$. Therefore, for a multiclass problem, the KL divergence can be expressed as:

$$D_{\text{KL}}(q\|\hat{p}) = \sum_j q_j \log(\frac{q_j}{\hat{p}_j}) =$$

$$= \sum_j q_j \log(q_j) - \sum_j q_j \log(\hat{p}_j); \Rightarrow \tag{5}$$

$$\Rightarrow D_{\text{KL}}(q\|\hat{p}) = -\mathbb{H}[q] + \mathcal{L}_{\text{c.e.}}(f);$$

where $\mathcal{L}_{\text{c.e.}}(f)$ is the cross-entropy loss, which forms an upper bond in the KL divergence:

$$\mathcal{L}_{\text{c.e.}}(f) = D_{\text{KL}}(q\|\hat{p}) + \mathbb{H}[q]; \Rightarrow$$

$$\Rightarrow \mathcal{L}_{\text{c.e.}}(f) \geq D_{\text{KL}}(q\|\hat{p}); \tag{6}$$

To simplify, we consider the case of the first selected epochs where $t_i \leftarrow y_i = 1$. Let $t_i \in q$, be the target distribution. If we take only one instance of m number of instances, i.e. $m = 1$, the loss function can be written as:

$$\mathcal{L}(f) = -\left[t_y(1 - \hat{p}_y)^\gamma \log \hat{p}_y + \alpha_{\text{dynamic}}(1 - t_y)(1 - \hat{p}_y)^\gamma \log \hat{p}_{idk}\right], \tag{7}$$

where the subscript $y$ denotes the values associated with the ground truth class.

Using Bernoulli's inequality, which states that $(1 - x)^\alpha \geq 1 - \alpha x$, if $0 \leq x \leq 1$ and $\alpha \geq 0$, as $\forall \gamma \geq 1$ and the $\hat{p}_y \in [0, 1]$, then we get:

$$\mathcal{L}(f) = -(1 - \hat{p}_y)^\gamma [t_y \log \hat{p}_y + \alpha_{\text{dynamic}}(1 - t_y) \log \hat{p}_{idk}]$$

$$\geq -(1 - \gamma \hat{p}_y)[t_y \log \hat{p}_y + \alpha_{\text{dynamic}}(1 - t_y) \log \hat{p}_{idk}]$$

$$= \gamma \hat{p}_y t_y \log \hat{p}_y - t_y \log \hat{p}_y + \gamma \hat{p}_y \alpha_{\text{dynamic}}(1 - t_y) \log \hat{p}_{idk} - \alpha_{\text{dynamic}}(1 - t_y) \log \hat{p}_{idk}$$

$$= -\gamma \mathbb{H}[\hat{p}] + \mathcal{L}_{\text{c.e.}}(f) + \alpha_{\text{dynamic}} \triangle_{reg} \tag{8}$$

$$= -\gamma \mathbb{H}[\hat{p}] + D_{\text{KL}}(q\|\hat{p}) + \mathbb{H}[q] + \alpha_{\text{dynamic}} \triangle_{reg};$$

$$\text{where } \triangle_{reg} = (1 - t_y)[\gamma \hat{p}_y \log \hat{p}_{idk} - \log \hat{p}_{idk}];$$

Therefore:

$$\mathcal{L}(f) \geq D_{\text{KL}}(q\|\hat{p}) + \mathbb{H}[q] - \gamma \mathbb{H}[\hat{p}] + \alpha_{\text{dynamic}} \triangle_{reg}; \tag{9}$$

where $\mathbb{H}[q]$ is a constant.

Thus, this new loss improves calibration by minimizing the KL divergence, maximizing the entropy depending on the weight of $\gamma$ (which smooths the learned distributions), and adding an extra regularization term (which might help to avoid overfitting) which maximises the uncertainty when the prediction is incorrect.

### C.2  SOCRATES LOSS REGULARIZES THE WEIGHTS OF THE NETWORK

Let $\mathcal{L}_{\text{c.e.}}(f)$ be cross-entropy loss, and $\mathcal{L}(f)$ be Socrates loss. The gradients of the neural network trained with $\mathcal{L}(f)$ are smaller than the ones trained with $\mathcal{L}_{\text{c.e.}}(f)$ when perfect confidence is reached and the model could start overfitting and subsequently be miscalibrated, i.e.,

$$\|\frac{\partial \mathcal{L}(f)}{\partial w}\| \leq \|\frac{\partial \mathcal{L}_{\text{c.e.}}(f)}{\partial w}\|. \tag{10}$$

This behaviour shows that Socrates loss acts as a regularizer when the model is sufficiently confident, avoiding miscalibration and overfitting.

**Proof:** To simplify, we consider the case of the first selected epochs where $t_i \leftarrow y_i = 1$. If we take only one instance from $m$ instances, i.e. $m = 1$, the Socrates loss function can be written as:

$$\mathcal{L}(f) = -\left[t_y(1 - \hat{p}_y)^\gamma \log \hat{p}_y + \alpha_{\text{dynamic}}(1 - t_y)(1 - \hat{p}_y)^\gamma \log \hat{p}_{idk}\right]. \tag{11}$$

The gradient with respect to the parameters of the last linear layer can be decomposed with the chain rule:

$$\frac{\partial \mathcal{L}(f)}{\partial w} = \frac{\partial \mathcal{L}(f)}{\partial \hat{p}_y} \frac{\partial \hat{p}_y}{\partial z} \frac{\partial z}{\partial w}.$$

$$\text{where } \frac{\partial \mathcal{L}(f)}{\partial \hat{p}_y} = \gamma(1 - \hat{p}_y)^{\gamma-1} t_y \log \hat{p}_y - (1 - \hat{p}_y)^\gamma \frac{t_y}{\hat{p}_y} + \tag{12}$$

$$+\gamma(1 - \hat{p}_y)^{\gamma-1}\alpha_{\text{dynamic}}(1 - t_y)\log \hat{p}_{idk} - (1 - \hat{p}_y)^\gamma \alpha_{\text{dynamic}}(1 - t_y)\frac{1}{\hat{p}_{idk}}.$$

On the other hand, cross-entropy loss can be written as:

$$\mathcal{L}_{\text{c.e.}}(f) = -t_y \log \hat{p}_y. \tag{13}$$

Where the gradient using the chain rule is:

$$\frac{\partial \mathcal{L}_{\text{c.e.}}(f)}{\partial w} = \frac{\partial \mathcal{L}_{\text{c.e.}}(f)}{\partial \hat{p}_y} \frac{\partial \hat{p}_y}{\partial z} \frac{\partial z}{\partial w}.$$

$$\text{where } \frac{\partial \mathcal{L}_{\text{c.e.}}(f)}{\partial \hat{p}_y} = -\frac{t_y}{\hat{p}_y} \tag{14}$$

Then, we can observe that the gradient of cross-entropy is a component of the gradient of Socrates:

$$\frac{\partial \mathcal{L}(f)}{\partial \hat{p}_y} = \frac{\partial \mathcal{L}_{\text{c.e.}}(f)}{\partial \hat{p}_y}\left[(1 - \hat{p}_y)^\gamma - \gamma \hat{p}_y(1 - \hat{p}_y)^{\gamma-1}\log \hat{p}_y\right] + \tag{15}$$

$$+\gamma(1 - \hat{p}_y)^{\gamma-1}\alpha_{\text{dynamic}}(1 - t_y)\log \hat{p}_{idk} - (1 - \hat{p}_y)^\gamma \alpha_{\text{dynamic}}(1 - t_y)\frac{1}{\hat{p}_{idk}}.$$

If $g(\hat{p}_y, \gamma) = (1 - \hat{p}_y)^\gamma - \gamma \hat{p}_y(1 - \hat{p}_y)^{\gamma-1}\log \hat{p}_y$ is a regularizer of the cross-entropy; and $r(t_y, \alpha_{\text{dynamic}}, \hat{p}_y, \hat{p}_{idk}) = \gamma(1 - \hat{p}_y)^{\gamma-1}\alpha_{\text{dynamic}}(1 - t_y)\log \hat{p}_{idk} - (1 - \hat{p}_y)^\gamma \alpha_{\text{dynamic}}(1 - t_y)\frac{1}{\hat{p}_{idk}}$ is highly affected by the idk class, which adds a small penalty $r(t_y, \alpha_{\text{dynamic}}, \hat{p}_y, \hat{p}_{idk}) \in [0, 1]$, then:

$$\frac{\partial \mathcal{L}(f)}{\partial \hat{p}_y} = \frac{\partial \mathcal{L}_{\text{c.e.}}(f)}{\partial \hat{p}_y} g(\hat{p}_y, \gamma) + r(t_y, \alpha_{\text{dynamic}}, \hat{p}_y, \hat{p}_{idk}). \tag{16}$$

When confidence is high, and the model could start being overfitted and miscalibrated, the value of $g(\hat{p}_y, \gamma) \in [0, 1]$. In that case:

$$||\frac{\partial \mathcal{L}(f)}{\partial \hat{p}_y}|| \leq ||\frac{\partial \mathcal{L}_{\text{c.e.}}(f)}{\partial \hat{p}_y}|| \implies ||\frac{\partial \mathcal{L}(f)}{\partial w}|| \leq ||\frac{\partial \mathcal{L}_{\text{c.e.}}(f)}{\partial w}|| \tag{17}$$

This demonstrates that the gradients of a model associated with the Socrates loss are smaller than those associated with the cross-entropy loss when perfect confidence is reached. Therefore, the Socrates loss acts as a regularizer with a penalty associated with the unknown knowledge of the classifier, avoiding overfitting, and subsequently miscalibration.

## D  MODEL REPRODUCIBILITY

### D.1  COMPUTE

The experiments were conducted on a shared supercomputer (Nvidia A100 80Gb SXM4 GPU). We consider it inequitable to provide specific time allocations for each method due to the nature of a shared supercomputer, where training durations vary based on resource availability. To ensure fair results, five different seeds were employed for each method, case, dataset, and architecture. The list of seeds to replicate results can be found in Table 3.

With certain methods and seeds, the training failed to achieve high accuracies, remaining stuck from the start at levels close to 10 and 20%. For SVHN, Focal loss could not train (converge) with VGG-16 when using seed 403. Additionally for SVHN, SAT and CCL-SC failed to train with VGG-16 using seed 303, and CCL-SC failed with VGG-16 for seeds 402, 403, 404, 405, and 409. For Food-101, SAT was unable to train with VGG-16 and ResNet-34 for any $E_{(s)} = 0$ seed. In contrast, Socrates successfully trained under all conditions.

Table 3: Seeds for results replication

| Dataset | Es | Seeds VGG-16 | Seeds ResNet-34 |
|---|---|---|---|
| CIFAR-10 and CIFAR-10C | 150 | 301, 302, 303, 304, 309 | 305, 306, 307, 308, 309 |
| | 0 | 401, 402, 403, 404, 409 | 405, 406, 407, 408, 409 |
| CIFAR-100 and CIFAR-100C | 150 | 301, 302, 303, 304, 309 | 305, 306, 307, 308, 309 |
| | 0 | 401, 402, 403, 404, 409 | 405, 406, 407, 408, 409 |
| SVHN | 150 | 301, 302, 303, 304, 309 | 305, 306, 307, 308, 309 |
| | 0 | 401, 402, 403, 404, 409 | 405, 406, 407, 408, 409 |
| Food-101 | 150 | 301, 312, 313, 314, 319 | 311, 312, 313, 314, 319 |
| | 0 | 401, 412, 413, 414, 419 | 411, 412, 413, 414, 419 |

### D.2 HYPERPARAMETERS

To conduct the experiments, we adapted the publicly available official implementation of Self-Adaptative Training, which was adapted from DeepGamblers (Liu et al., 2019). The hyperparameter values do not vary from the SAT implementation to ensure a fair comparison.

All models were trained for 300 epochs without early stopping. CIFAR-10, CIFAR-100, and SVHN were trained with a mini-batch size of 128 for training and 200 for testing. Due to resource limitations, Food-101 was trained with a mini-batch size of 128 for both training and testing.

The models were trained using SGD with an initial learning rate of 0.1 and a momentum of 0.9. The learning rate was reduced by 0.5 every 25 epochs. Weight decay was set to 0.0005.

For SAT and Socrates, an additional class (the idk class) was added, and the momentum of the loss was set to 0.9.

For Focal and Socrates, the gamma of the losses was set to 2, and alpha was set to 1.

The Selective Classification problem was evaluated with the coverage levels: 100, 98, 97, 95, 90, 85, 80, 75, 70, 60, 50, 40, 30, 20, and 10.

### D.3 DATASETS

As indicated by Feng et al. (2023), SAT was tested on easy-to-classify datasets. Therefore, for a more comprehensive analysis, we have selected a wide range of datasets with variable degrees of complexity. First, we chose the easy-to-classify CIFAR-10 and SVHN datasets. Although improvements may be less apparent with these toy datasets, the drawbacks of the methods could become more noticeable. To increase the challenge, we included the CIFAR-100 and Food-101 datasets. In particular, Food-101 serves as a good example of a real-world dataset, testing the reliability aspect of our new method. To further explore reliability, we tested the robustness of CIFAR-10 and CIFAR-100 using the CIFAR-10C and CIFAR-100C datasets as test sets.

The Street View House Number (SVHN) (Netzer et al., 2011) contains 73257 training and 26032 evaluation real-world small images of 32x32x3 with 10 classes. CIFAR-10 (Krizhevsky, 2009) comprises 50000 training and 10000 evaluation small images of 32x32x3 with 10 classes. CIFAR-100 (Krizhevsky, 2009) is like CIFAR-10 with 50000 training and 10000 evaluation small images of 32x32x3 but with 100 classes. CIFAR-10C (Hendrycks & Dietterich, 2019) comprises 50000 test small images of 32x32x3 with 10 classes created using the 10000 evaluation images using 5 different levels of corruption. CIFAR-100C (Hendrycks & Dietterich, 2019) similar to CIFAR-10C but with 100 classes. Food-101(Bossard et al., 2014) constitutes 75750 training and 25250 evaluation images of 224x224x3 with 101 food classes.

# E   SELECTIVE CLASSIFICATION ERROR RATE RESULTS AND ECE IN THE 300 EPOCH

As indicated in Subsection 6.2.1 of the main paper, the goal of this study is to produce calibrated selective classifiers that aim to achieve Selective Classification results similar to or better than those of SAT and CCL-SC, while ensuring well-calibrated confidence levels. The Selective Classification error rates achieved are comparable to or superior to those reached by SAT and CCL-SC. Notably, for the challenging CIFAR-100 and Food-101 datasets, Socrates significantly outperforms the Selective Classification error rates achieved by SAT. In this framework, once the model has been trained, it is insufficient to evaluate only the Selective Classification error rate without also considering metrics such as the ECE and accuracy.

The mean and standard deviation of the ECE values for the 300 epoch can be seen in Table 4 for the VGG-16 architecture and in Table 5 for the ResNet-34 architecture. The Selective Classification Error rates can be seen in Table 6 for the VGG-16 architecture, and in Table 7 for the Resnet-34 architecture.

Table 4: **ECE** values in a range of $[0, 1]$ and **accuracy** (acc) values ($100\%$) on the 300 epoch with the CIFAR-10, CIFAR-100, SVHN, Food-101, CIFAR-10C, and CIFAR-100 datasets with mean and standard deviation for trainings with **VGG-16 architecture**. A notable improvement can be seen in Food-101 dataset. Underline indicate the overall best performance, while bold highlight the best performance in each case.

| Dataset | Coverage | *end-to-end* case | | *first-epochs* case | | |
| | | Socrates (ours) | SAT | Socrates + Focal | CCL-SC + c.e | SAT + c.e |
|---|---|---|---|---|---|---|
| CIFAR-10 | Acc Train | **97.47 ± 0.11** | 94.33 ± 3.65 | 97.50 ± 0.28 | ***97.79 ± 0.11*** | 95.60 ± 3.80 |
| | ECE Train | **0.003 ± 0.0004** | 0.03 ± 0.01 | ***0.004 ± 0.0005*** | 0.007 ± 0.001 | 0.04 ± 0.008 |
| | Acc val | **99.53 ± 0.03** | 97.37 ± 2.36 | 99.81 ± 0.06 | ***99.93 ± 0.02*** | 98.50 ± 2.73 |
| | ECE Val | **0.003 ± 0.0003** | 0.02 ± 0.01 | ***0.004 ± 0.001*** | 0.005 ± 0.001 | 0.02 ± 0.001 |
| | ECE Test | 0.04 ± 0.002 | **0.02 ± 0.005** | 0.04 ± 0.002 | *0.04 ± 0.001* | **0.02 ± 0.01** |
| CIFAR-100 | Acc Train | **83.64 ± 0.30** | 50.84 ± 1.09 | 84.18 ± 0.56 | ***85.59 ± 0.54*** | 69.93 ± 0.47 |
| | ECE Train | **0.015 ± 0.001** | 0.37 ± 0.015 | 0.03 ± 0.002 | ***0.017 ± 0.001*** | 0.10 ± 0.002 |
| | Acc Val | **94.06 ± 0.18** | 57.49 ± 1.37 | 95.74 ± 0.35 | ***97.04 ± 0.28*** | 88.38 ± 0.53 |
| | ECE Val | **0.006 ± 0.001** | 0.35 ± 0.02 | ***0.01 ± 0.001*** | 0.02 ± 0.001 | 0.05 ± 0.002 |
| | ECE Test | **0.126 ± 0.004** | 0.41 ± 0.01 | ***0.12 ± 0.003*** | 0.13 ± 0.002 | 0.14 ± 0.002 |
| SVHN | Acc Train | **98.59 ± 0.1** | 97.78 ± 0.04 | ***98.69 ± 0.1*** | 82.79 ± 35.67 | 78.81 ± 44.06 |
| | ECE Train | **0.003 ± 0.0001** | 0.01 ± 0.0001 | ***0.003 ± 0.0002*** | 0.004 ± 0.003 | 0.18 ± 0.36 |
| | Acc Val | **99.42 ± 0.04** | 98.82 ± 0.03 | ***99.64 ± 0.08*** | 85.57 ± 36.11 | 79.62 ± 44.51 |
| | ECE Val | **0.002 ± 0.0002** | 0.008 ± 0.0005 | ***0.002 ± 0.0001*** | 0.003 ± 0.002 | 0.17 ± 0.37 |
| | ECE Test | 0.013 ± 0.001 | **0.007 ± 0.001** | 0.015 ± 0.001 | ***0.012 ± 0.01*** | 0.17 ± 0.37 |
| Food-101 | Acc Train | **66.58 ± 0.86** | 21.94 ± 1.78 | 66.51 ± 0.34 | ***68.98 ± 0.44*** | 40.06 ± 0.68 |
| | ECE Train | **0.04 ± 0.002** | 0.66 ± 0.03 | ***0.04 ± 0.003*** | 0.04 ± 0.003 | 0.31 ± 0.007 |
| | Acc Val | **74.48 ± 0.74** | 26.52 ± 2.15 | 74.60 ± 0.20 | ***75.58 ± 0.35*** | 55.08 ± 0.58 |
| | ECE Val | **0.017 ± 0.002** | 0.61 ± 0.03 | ***0.025 ± 0.004*** | 0.027 ± 0.004 | 0.20 ± 0.005 |
| | ECE Test | **0.016 ± 0.003** | 0.61 ± 0.03 | 0.017 ± 0.002 | ***0.011 ± 0.0003*** | 0.20 ± 0.01 |
| CIFAR-10C | ECE Test | 0.145 ± 0.002 | **0.114 ± 0.003** | *0.154 ± 0.003* | 0.156 ± 0.01 | **0.11 ± 0.004** |
| CIFAR-100C | ECE Test | **0.24 ± 0.003** | 0.51 ± 0.01 | ***0.24 ± 0.004*** | 0.25 ± 0.12 | 0.28 ± 0.001 |

Table 5: **ECE** values in a range of $[0, 1]$ and **accuracy** (acc) values (100%) on the 300 epoch with the CIFAR-10, CIFAR-100, SVHN, Food-101, CIFAR-10C, and CIFAR-100 datasets with mean and standard deviation for trainings with **ResNet-34 architecture**. A notable improvement can be seen in Food-101 dataset. Underline indicate the overall best performance, while bold highlight the best performance in each case.

| Dataset | Coverage | *end-to-end* case | | | *first-epochs* case | |
|---|---|---|---|---|---|---|
| | | Socrates (ours) | SAT | Socrates + Focal | CCL-SC + c.e | SAT + c.e |
| CIFAR-10 | Acc Train | **99.998 $\pm$ 0.001** | 99.98 $\pm$ 0.02 | ***99.999 $\pm$ 0.001*** | 99.998 $\pm$ 0.002 | 99.994 $\pm$ 0.01 |
| | ECE Train | **0.001 $\pm$ 0.0001** | 0.002 $\pm$ 0.0002 | ***0.001 $\pm$ 0.00004*** | 0.001 $\pm$ 0.002 | 0.003 $\pm$ 0.0003 |
| | Acc Val | **100 $\pm$ 0** | 99.98 $\pm$ 0.003 | ***100 $\pm$ 0*** | **100 $\pm$ 0** | 99.99 $\pm$ 0.01 |
| | ECE Val | **0.00077 $\pm$ 0.0001** | 0.002 $\pm$ 0.0002 | ***0.00076 $\pm$ 0.0001*** | 0.0008 $\pm$ 0.0001 | 0.002 $\pm$ 0.0004 |
| | ECE Test | **0.033 $\pm$ 0.0009** | 0.033 $\pm$ 0.001 | 0.037 $\pm$ 0.003 | *0.034 $\pm$ 0.002* | **0.032 $\pm$ 0.003** |
| CIFAR-100 | Acc Train | **99.983 $\pm$ 0.006** | 99.26 $\pm$ 0.07 | 99.97 $\pm$ 0.005 | ***99.9827 $\pm$ 0.007*** | 99.96 $\pm$ 0.01 |
| | ECE Train | 0.0061 $\pm$ 0.0006 | **0.0058 $\pm$ 0.001** | 0.0064 $\pm$ 0.0002 | ***0.006 $\pm$ 0.0004*** | 0.01 $\pm$ 0.001 |
| | Acc Val | **99.9813 $\pm$ 0.005** | 99.27 $\pm$ 0.07 | ***99.984 $\pm$ 0.003*** | 99.9836 $\pm$ 0.003 | 99.97 $\pm$ 0.01 |
| | ECE Val | 0.002 $\pm$ 0.0002 | **0.001 $\pm$ 0.0003** | 0.0025 $\pm$ 0.0002 | ***0.0002 $\pm$ 0.0001*** | 0.006 $\pm$ 0.0004 |
| | ECE Test | **0.067 $\pm$ 0.01** | 0.07 $\pm$ 0.02 | 0.07 $\pm$ 0.01 | ***0.064 $\pm$ 0.01*** | 0.065 $\pm$ 0.01 |
| SVHN | Acc Train | **99.99 $\pm$ 0.002** | 99.86 $\pm$ 0.02 | ***99.9948 $\pm$ 0.002*** | 99.99 $\pm$ 0.004 | 99.99 $\pm$ 0.002 |
| | ECE Train | **0.00102 $\pm$ 0.0001** | 0.002 $\pm$ 0.0004 | ***0.00104 $\pm$ 0.0001*** | 0.003 $\pm$ 0.001 | 0.002 $\pm$ 0.0002 |
| | Acc Val | **99.997 $\pm$ 0.001** | 99.86 $\pm$ 0.02 | 99.9957 $\pm$ 0.001 | ***99.996 $\pm$ 0.001*** | 99.995 $\pm$ 0.002 |
| | ECE Val | 0.0008 $\pm$ 0.0001 | 0.001 $\pm$ 0.0003 | 0.00067 $\pm$ 0.0001 | ***0.00065 $\pm$ 0.0001*** | 0.002 $\pm$ 0.0003 |
| | ECE Test | **0.019 $\pm$ 0.001** | **0.019 $\pm$ 0.001** | 0.021 $\pm$ 0.001 | *0.02 $\pm$ 0.001* | **0.018 $\pm$ 0.001** |
| Food-101 | Acc Train | **95.78 $\pm$ 0.31** | 0 $\pm$ 0 | 82.38 $\pm$ 29.08 | ***95.52 $\pm$ 0.27*** | 88.85 $\pm$ 2.33 |
| | ECE Train | **0.021 $\pm$ 0.002** | 1 $\pm$ 0 | 0.026 $\pm$ 0.002 | ***0.023 $\pm$ 0.002*** | 0.08 $\pm$ 0.004 |
| | Acc Val | **98.05 $\pm$ 0.37** | 0 $\pm$ 0 | 82.95 $\pm$ 33.27 | ***97.57 $\pm$ 0.69*** | 92.19 $\pm$ 2.51 |
| | ECE Val | **0.023 $\pm$ 0.004** | 1 $\pm$ 0 | 0.04 $\pm$ 0.03 | ***0.026 $\pm$ 0.003*** | 0.06 $\pm$ 0.005 |
| | ECE Test | **0.067 $\pm$ 0.001** | 1 $\pm$ 0 | *0.07 $\pm$ 0.02* | 0.078 $\pm$ 0.002 | 0.09 $\pm$ 0.01 |
| CIFAR10C | ECE Test | 0.19 $\pm$ 0.01 | **0.18 $\pm$ 0.01** | *0.19 $\pm$ 0.01* | 0.18 $\pm$ 0.01 | **0.17 $\pm$ 0.01** |
| CIFAR100C | ECE Test | **0.16 $\pm$ 0.04** | 0.18 $\pm$ 0.04 | ***0.17 $\pm$ 0.03*** | 0.18 $\pm$ 0.02 | 0.20 $\pm$ 0.01 |

Table 6: **Selective Classification error rate** % on the 300 epoch with the CIFAR-10, CIFAR-100, SVHN, Food-101, CIFAR-10C, and CIFAR-100 datasets for various coverage rates % with mean and standard deviation for trainings with **VGG-16 architecture**. A notable improvement can be seen in Food-101 dataset. CCL-SC was not able to perform correctly for SVHN, SAT was not able for Food-101. Underline indicate the overall best performance, while bold highlight the best performance in each case.

| Dataset | Coverage | end-to-end case | | first-epochs case | | |
| | | Socrates (ours) | SAT | Socrates + Focal | CCL-SC + c.e | SAT + c.e |
|---|---|---|---|---|---|---|
| CIFAR-10 | 100 | **6.44 ± 0.18** | 7.08 ± 1.07 | 6.67 ± 0.19 | _**6.38 ± 0.14**_ | 6.87 ± 1.08 |
| | 95 | **4.14 ± 0.12** | 4.78 ± 0.98 | 4.45 ± 0.14 | _**4.02 ± 0.14**_ | 4.58 ± 1.12 |
| | 90 | **2.43 ± 0.09** | 3.01 ± 0.88 | 2.76 ± 0.13 | _**2.36 ± 0.13**_ | 2.92 ± 1.01 |
| | 85 | **1.48 ± 0.11** | 1.82 ± 0.65 | 1.64 ± 0.20 | _**1.47 ± 0.16**_ | 1.75 ± 0.74 |
| | 80 | **0.85 ± 0.03** | 1.12 ± 0.51 | _**1.05 ± 0.11**_ | 1.18 ± 0.25 | 1.05 ± 0.46 |
| | 75 | _**0.52 ± 0.03**_ | 0.67 ± 0.32 | 0.68 ± 0.07 | 1.05 ± 0.19 | **0.61 ± 0.27** |
| | 70 | _**0.38 ± 0.04**_ | 0.43 ± 0.24 | _0.51 ± 0.05_ | 0.91 ± 0.11 | **0.42 ± 0.20** |
| CIFAR-100 | 100 | **28.04 ± 0.24** | 47.74 ± 1.27 | 28.08 ± 0.24 | _28.01 ± 0.27_ | _**28.00 ± 0.06**_ |
| | 95 | **25.45 ± 0.32** | 45.03 ± 1.33 | 25.45 ± 0.32 | 25.49 ± 0.31 | _**25.16 ± 0.08**_ |
| | 90 | **22.85 ± 0.30** | 42.07 ± 1.38 | 23.07 ± 0.36 | 22.95 ± 0.28 | _**22.57 ± 0.07**_ |
| | 85 | **20.23 ± 0.30** | 38.89 ± 1.45 | 20.76 ± 0.37 | _20.37 ± 0.30_ | _**20.06 ± 0.08**_ |
| | 80 | **17.70 ± 0.23** | 35.50 ± 1.43 | 18.34 ± 0.37 | _17.79 ± 0.34_ | _**17.65 ± 0.09**_ |
| | 75 | **15.25 ± 0.28** | 31.81 ± 1.48 | 15.85 ± 0.37 | _15.27 ± 0.35_ | _**15.20 ± 0.12**_ |
| | 70 | **12.95 ± 0.29** | 28.00 ± 1.42 | 13.62 ± 0.49 | _12.89 ± 0.21_ | _**12.86 ± 0.14**_ |
| SVHN | 100 | 2.72 ± 0.07 | _**2.65 ± 0.04**_ | _**2.80 ± 0.03**_ | 18.29 ± 34.73 | 18.22 ± 34.77 |
| | 95 | 1.15 ± 0.04 | _**1.04 ± 0.02**_ | _**1.20 ± 0.08**_ | 16.99 ± 35.46 | 16.89 ± 35.51 |
| | 90 | 0.74 ± 0.05 | _**0.61 ± 0.05**_ | _**0.80 ± 0.05**_ | 16.76 ± 35.58 | 16.57 ± 35.69 |
| | 85 | 0.62 ± 0.02 | _**0.45 ± 0.04**_ | _**0.62 ± 0.05**_ | 16.70 ± 35.62 | 16.44 ± 35.76 |
| | 80 | 0.55 ± 0.03 | _**0.38 ± 0.02**_ | _**0.54 ± 0.05**_ | 16.66 ± 35.64 | 16.39 ± 35.79 |
| | 75 | 0.49 ± 0.05 | _**0.33 ± 0.02**_ | _**0.51 ± 0.03**_ | 16.64 ± 35.65 | 16.35 ± 35.81 |
| | 70 | 0.45 ± 0.04 | _**0.30 ± 0.01**_ | _**0.48 ± 0.02**_ | 16.62 ± 35.66 | 16.33 ± 35.82 |
| Food-101 | 100 | **26.93 ± 0.52** | 68.23 ± 2.19 | _**27.08 ± 0.25**_ | 27.18 ± 0.19 | 29.00 ± 0.27 |
| | 95 | **24.62 ± 0.54** | 66.56 ± 2.31 | 24.78 ± 0.22 | _**24.62 ± 0.19**_ | 26.63 ± 0.23 |
| | 90 | **22.19 ± 0.50** | 64.74 ± 2.43 | 22.40 ± 0.27 | _**22.04 ± 0.18**_ | 24.29 ± 0.27 |
| | 85 | **19.75 ± 0.52** | 62.72 ± 2.56 | 20.00 ± 0.23 | _**19.38 ± 0.20**_ | 21.89 ± 0.25 |
| | 80 | **17.16 ± 0.60** | 60.48 ± 2.70 | 17.59 ± 0.22 | _**16.75 ± 0.17**_ | 19.43 ± 0.25 |
| | 75 | **14.64 ± 0.63** | 57.97 ± 2.86 | 15.18 ± 0.19 | _**14.19 ± 0.20**_ | 17.06 ± 0.21 |
| | 70 | **12.16 ± 0.56** | 55.13 ± 3.04 | 12.87 ± 0.24 | _**11.67 ± 0.24**_ | 14.66 ± 0.29 |
| CIFAR10C | 100 | 21.91 ± 0.24 | _**21.67 ± 0.22**_ | 22.43 ± 0.35 | _**22.05 ± 0.57**_ | 22.53 ± 1.95 |
| | 95 | 19.29 ± 0.26 | _**19.10 ± 0.25**_ | 19.95 ± 0.42 | _**19.67 ± 0.59**_ | 20.01 ± 2.04 |
| | 90 | 16.89 ± 0.28 | _**16.71 ± 0.25**_ | 17.65 ± 0.45 | _**17.31 ± 0.59**_ | 17.65 ± 2.11 |
| | 85 | 14.61 ± 0.28 | _**14.44 ± 0.24**_ | 15.42 ± 0.46 | _**14.98 ± 0.59**_ | 15.40 ± 2.16 |
| | 80 | 12.40 ± 0.27 | _**12.30 ± 0.23**_ | 13.23 ± 0.43 | _**12.70 ± 0.58**_ | 13.26 ± 2.16 |
| | 75 | 10.29 ± 0.27 | _**10.27 ± 0.23**_ | 11.12 ± 0.40 | _**10.55 ± 0.55**_ | 11.23 ± 2.16 |
| | 70 | **8.33 ± 0.27** | 8.41 ± 0.22 | 9.17 ± 0.35 | _**8.61 ± 0.52**_ | 9.34 ± 2.09 |
| CIFAR100C | 100 | **49.57 ± 0.14** | 60.79 ± 0.78 | 49.68 ± 0.09 | 49.43 ± 0.31 | _**49.03 ± 0.20**_ |
| | 95 | **47.66 ± 0.13** | 58.83 ± 0.81 | 47.59 ± 0.14 | 47.56 ± 0.32 | _**46.90 ± 0.21**_ |
| | 90 | **45.63 ± 0.13** | 56.77 ± 0.84 | _45.47 ± 0.18_ | 45.57 ± 0.34 | _**44.70 ± 0.21**_ |
| | 85 | **43.47 ± 0.12** | 54.58 ± 0.85 | _43.31 ± 0.23_ | 43.47 ± 0.34 | _**42.40 ± 0.22**_ |
| | 80 | **41.15 ± 0.12** | 52.23 ± 0.87 | _41.09 ± 0.27_ | 41.24 ± 0.36 | _**40.00 ± 0.21**_ |
| | 75 | **38.66 ± 0.13** | 49.70 ± 0.89 | _38.73 ± 0.30_ | 38.85 ± 0.36 | _**37.49 ± 0.21**_ |
| | 70 | **36.01 ± 0.12** | 46.94 ± 0.91 | _36.22 ± 0.34_ | 36.30 ± 0.36 | _**34.86 ± 0.22**_ |

Table 7: **Selective Classification error rate** % on the 300 epoch with the CIFAR-10, CIFAR-100, SVHN, Food-101, CIFAR-10C, and CIFAR-100 datasets for various coverage rates % with mean and standard deviation for trainings with **ResNet-34 architecture**. A notable improvement can be seen in Food-101 dataset. SAT was not able to perform correctly for Food-101. Underline indicate the overall best performance, while bold highlight the best performance in each case.

| | | *end-to-end* case | | *first-epochs* case | | |
| Dataset | Coverage | Socrates (ours) | SAT | Socrates + Focal | CCL-SC + c.e | SAT + c.e |
| --- | --- | --- | --- | --- | --- | --- |
| CIFAR-10 | 100 | **4.95 ± 0.19** | 5.10 ± 0.32 | 5.15 ± 0.26 | *5.07 ± 0.10* | **4.97 ± 0.14** |
| | 95 | **2.71 ± 0.19** | 2.85 ± 0.29 | 2.95 ± 0.20 | *2.87 ± 0.11* | **2.84 ± 0.14** |
| | 90 | **1.46 ± 0.13** | 1.55 ± 0.24 | 1.65 ± 0.17 | ***1.53 ± 0.12*** | 1.57 ± 0.15 |
| | 85 | **0.81 ± 0.11** | 0.88 ± 0.09 | 1.08 ± 0.12 | *0.90 ± 0.11* | **0.90 ± 0.08** |
| | 80 | **0.56 ± 0.09** | 0.60 ± 0.09 | 0.88 ± 0.09 | *0.66 ± 0.06* | **0.60 ± 0.09** |
| | 75 | 0.46 ± 0.07 | **0.43 ± 0.11** | 0.79 ± 0.10 | *0.47 ± 0.03* | **0.44 ± 0.10** |
| | 70 | 0.40 ± 0.09 | **0.30 ± 0.08** | 0.73 ± 0.08 | *0.39 ± 0.04* | **0.36 ± 0.07** |
| CIFAR-100 | 100 | **22.74 ± 0.34** | 23.26 ± 0.50 | 23.19 ± 0.51 | *23.23 ± 0.67* | **22.85 ± 0.27** |
| | 95 | **20.24 ± 0.45** | 20.51 ± 0.51 | 20.48 ± 0.43 | *20.44 ± 0.68* | **20.06 ± 0.37** |
| | 90 | **17.62 ± 0.59** | 17.81 ± 0.43 | 18.05 ± 0.35 | *17.80 ± 0.63* | **17.53 ± 0.43** |
| | 85 | **15.15 ± 0.54** | 15.30 ± 0.46 | 15.71 ± 0.28 | *15.23 ± 0.64* | **15.23 ± 0.36** |
| | 80 | 12.85 ± 0.60 | **12.83 ± 0.44** | 13.62 ± 0.16 | *12.90 ± 0.70* | 13.03 ± 0.26 |
| | 75 | 10.74 ± 0.56 | **10.73 ± 0.39** | 11.68 ± 0.24 | ***10.69 ± 0.63*** | 11.09 ± 0.24 |
| | 70 | 8.73 ± 0.59 | **8.65 ± 0.40** | 9.84 ± 0.39 | ***8.67 ± 0.57*** | 9.19 ± 0.24 |
| SVHN | 100 | **2.66 ± 0.09** | 2.77 ± 0.08 | 2.78 ± 0.08 | *2.74 ± 0.06* | **2.73 ± 0.11** |
| | 95 | 1.02 ± 0.03 | **0.99 ± 0.06** | 1.11 ± 0.03 | ***1.03 ± 0.05*** | 1.04 ± 0.05 |
| | 90 | 0.65 ± 0.06 | **0.60 ± 0.03** | 0.76 ± 0.05 | *0.67 ± 0.04* | **0.65 ± 0.07** |
| | 85 | 0.55 ± 0.06 | **0.48 ± 0.03** | 0.67 ± 0.07 | *0.59 ± 0.07* | **0.54 ± 0.06** |
| | 80 | 0.52 ± 0.04 | **0.43 ± 0.03** | 0.60 ± 0.09 | *0.56 ± 0.08* | **0.48 ± 0.05** |
| | 75 | 0.48 ± 0.05 | **0.40 ± 0.03** | 0.56 ± 0.08 | *0.54 ± 0.08* | **0.44 ± 0.05** |
| | 70 | 0.46 ± 0.05 | **0.39 ± 0.03** | 0.53 ± 0.07 | *0.51 ± 0.07* | **0.43 ± 0.05** |
| Food-101 | 100 | **21.40 ± 0.79** | 100 ± 0.0 | 32.33 ± 22.32 | *22.77 ± 0.90* | **22.08 ± 0.75** |
| | 95 | **18.95 ± 0.80** | 100 ± 0.0 | 30.20 ± 23.10 | *20.09 ± 0.92* | **20.02 ± 0.74** |
| | 90 | **16.54 ± 0.75** | 100 ± 0.0 | 28.23 ± 23.92 | ***17.39 ± 0.91*** | 17.97 ± 0.74 |
| | 85 | **14.32 ± 0.74** | 100 ± 0.0 | 26.37 ± 23.92 | ***14.75 ± 0.92*** | 15.99 ± 0.72 |
| | 80 | **12.30 ± 0.78** | 100 ± 0.0 | 24.60 ± 25.11 | ***12.30 ± 0.94*** | 14.08 ± 0.67 |
| | 75 | **10.32 ± 0.68** | 100 ± 0.0 | 22.94 ± 25.57 | ***10.00 ± 0.81*** | 12.20 ± 0.64 |
| | 70 | **8.54 ± 0.62** | 100 ± 0.0 | 21.49 ± 25.97 | ***7.85 ± 0.70*** | 10.37 ± 0.60 |
| CIFAR10C | 100 | 24.64 ± 0.49 | **24.30 ± 0.89** | 24.50 ± 0.66 | ***24.28 ± 0.46*** | 24.73 ± 0.67 |
| | 95 | 22.08 ± 0.49 | **21.74 ± 0.92** | 21.97 ± 0.67 | ***21.93 ± 0.47*** | 22.17 ± 0.72 |
| | 90 | 19.65 ± 0.46 | **19.30 ± 0.92** | 19.57 ± 0.70 | ***19.57 ± 0.47*** | 19.73 ± 0.77 |
| | 85 | 17.27 ± 0.44 | **16.95 ± 0.89** | 17.24 ± 0.73 | ***17.18 ± 0.47*** | 17.36 ± 0.85 |
| | 80 | 14.95 ± 0.41 | **14.66 ± 0.85** | 14.95 ± 0.77 | ***14.83 ± 0.46*** | 15.05 ± 0.95 |
| | 75 | 12.68 ± 0.37 | **12.46 ± 0.78** | 12.74 ± 0.80 | ***12.53 ± 0.44*** | 12.80 ± 1.02 |
| | 70 | **10.56 ± 0.31** | 10.69 ± 1.08 | 10.68 ± 0.81 | *10.35 ± 0.39* | **10.22 ± 0.69** |
| CIFAR100C | 100 | **49.14 ± 0.32** | 49.29 ± 0.72 | 49.73 ± 0.59 | *49.04 ± 0.59* | **48.83 ± 0.24** |
| | 95 | 47.46 ± 0.30 | **47.38 ± 0.72** | 47.73 ± 0.63 | *47.11 ± 0.62* | **46.83 ± 0.28** |
| | 90 | 45.56 ± 0.29 | **45.38 ± 0.74** | 45.69 ± 0.67 | *45.06 ± 0.64* | **44.75 ± 0.31** |
| | 85 | 43.49 ± 0.29 | **43.24 ± 0.74** | 43.58 ± 0.71 | *42.92 ± 0.66* | **42.56 ± 0.33** |
| | 80 | 41.26 ± 0.28 | **40.95 ± 0.77** | 41.37 ± 0.75 | *40.66 ± 0.67* | **40.26 ± 0.33** |
| | 75 | 38.87 ± 0.28 | **38.48 ± 0.80** | 39.05 ± 0.80 | *38.28 ± 0.67* | **37.86 ± 0.33** |
| | 70 | 36.33 ± 0.26 | **35.83 ± 0.81** | 36.64 ± 0.85 | *35.76 ± 0.66* | **35.35 ± 0.34** |

## F  IS SELF-ADAPTIVE TRAINING LOSS A CALIBRATION LOSS? DETAILED ANALYSIS

**Overfitting:**  As stated by Mukhoti et al. (2020) and Guo et al. (2017), overfitting appears to be linked to miscalibration. Therefore, the first step towards evaluating SAT as a calibrator is to examine the accuracy and loss curves to reconfirm the alleviation of the overfitting issue (one of the claims of the SAT and Focal methods). These curves are presented in Figure 6 and 7. Focal loss consistently maintains the same trend and does not induce overfitting with any dataset or architecture, except for the Food-101 dataset, where minor overfitting occurs in the initial epochs. In contrast, SAT loss behaves differently and does not consistently prevent overfitting. SAT shows overfitting with the SVHN and Food-101 datasets with the VGG-16 architecture during the *first-epochs* case, and for the *end-to-end* case with the challenging CIFAR-100 and Food-101 datasets across both architectures. Furthermore, with the CIFAR-100 and Food-101 datasets with the VGG-16 architecture, the accuracy achieved with SAT loss is significantly lower than with Focal loss. For the *end-to-end* case with Food-101, and ResNet-34, SAT could not train. Although these observations suggest that SAT may not be an effective calibration loss, they do not provide definitive evidence of miscalibration, necessitating further analysis.

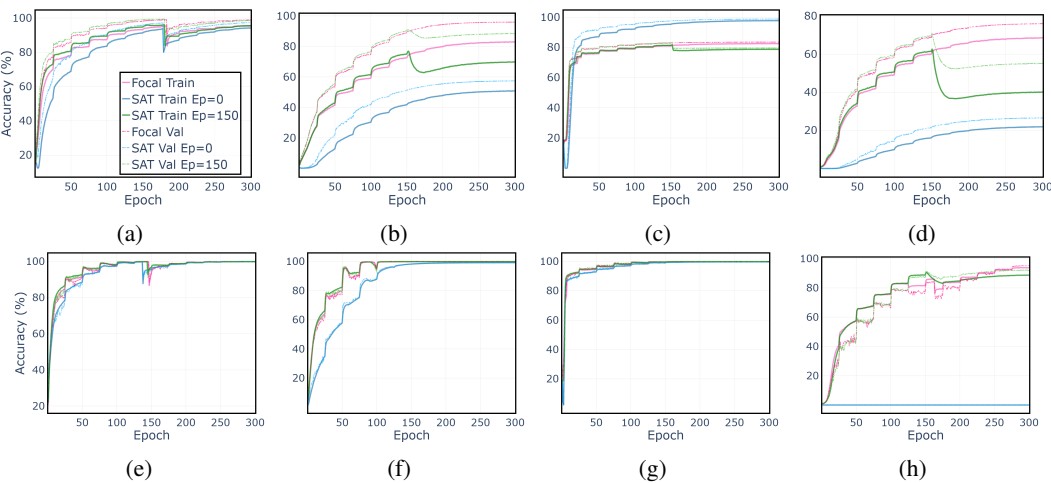

Figure 6: Accuracy curves of models trained on CIFAR-10 (a and e), CIFAR-100 (b and f) SVHN (c and g) and Food-101 (d and h) datasets using Focal and SAT (*first-epochs* and *end-to-end* cases) methods with VGG-16 (a, b, c, and d) and ResNet-34 (e, f, g, and h) architectures.

**Calibration Metrics:**  The second step towards the analysis of SAT as a calibration loss is to visualize specific calibration metrics. The snapshot of the ECE and MCE metrics in the reliability diagram of the last training epoch does not give enough insights to output calibration conclusions, instead, guided for the experimentation phase made in Mukhoti et al. (2020), the ECE and MCE values in each epoch of the training process produce noticeable insights. Therefore, the visualization starts analyzing the ECE value along the epochs. These ECE values along epochs curves can be seen in Figure 8.

The ECE along epochs curves of Focal loss exhibit a consistent downward trend across both architectures, except for VGG-16 with CIFAR-100 and Food-101 datasets where there is an increase in the initial epochs but in an acceptable ECE range. Regarding SAT, in the first-epochs case, ECE values for VGG-16 architectures rise significantly after 150 epochs, especially in the Food-101 dataset, but this increase is less noticeable for ResNet-34 architectures. In the end-to-end case, both architectures show high initial ECE values that gradually decrease, though VGG-16 has a particularly high ECE of around 0.9, compared to much lower values with focal loss. These observations suggest that SAT is less reliable as a calibrator compared to focal loss, which performs more consistently.

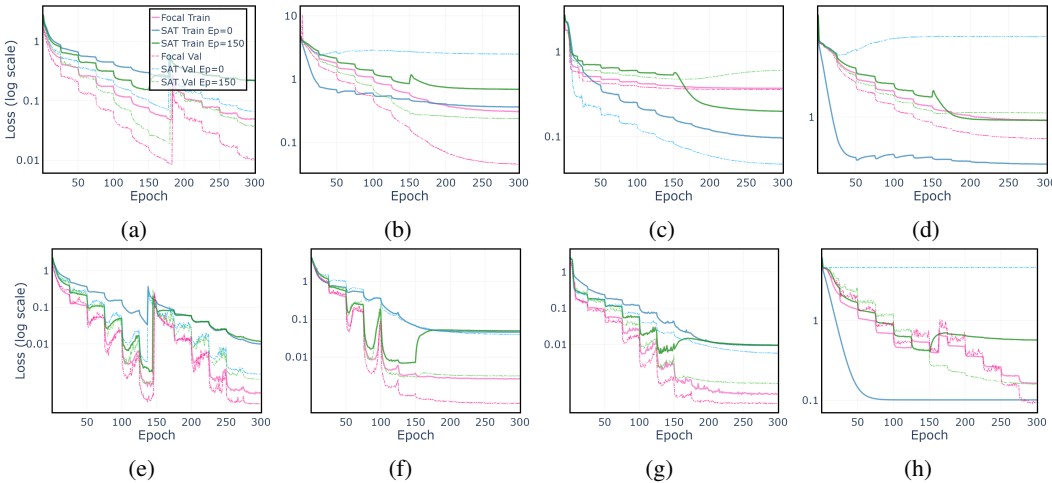

Figure 7: Loss curves of models trained on CIFAR-10 (a and e), CIFAR-100 (b and f), SVHN (c and g), and Food-101 (d and h) datasets using Focal and SAT (*first-epochs* and *end-to-end* cases) methods with VGG-16 (a, b, c, and d), and ResNet-34 (e, f, g, and h) architectures.

The MCE curves do not provide sufficient insights, as the MCE values are typically driven by only a few instances. The main claim is that the SAT exhibits distinct trends in both VGG-16 and ResNet-34 architectures compared to Focal loss, mirroring the observations made regarding ECE.

Therefore, a significant claim can be put forth: SAT loss does not seem to be a good loss for training calibrated models, and it appears detrimental when the goal is to train for a small number of epochs. It is well-known that the aim is not always to train for longer, as it depends on the dataset and the architecture. In this case, the SAT loss outputs calibrated confidences after a considerable amount of epochs, which may not be desirable in all cases.

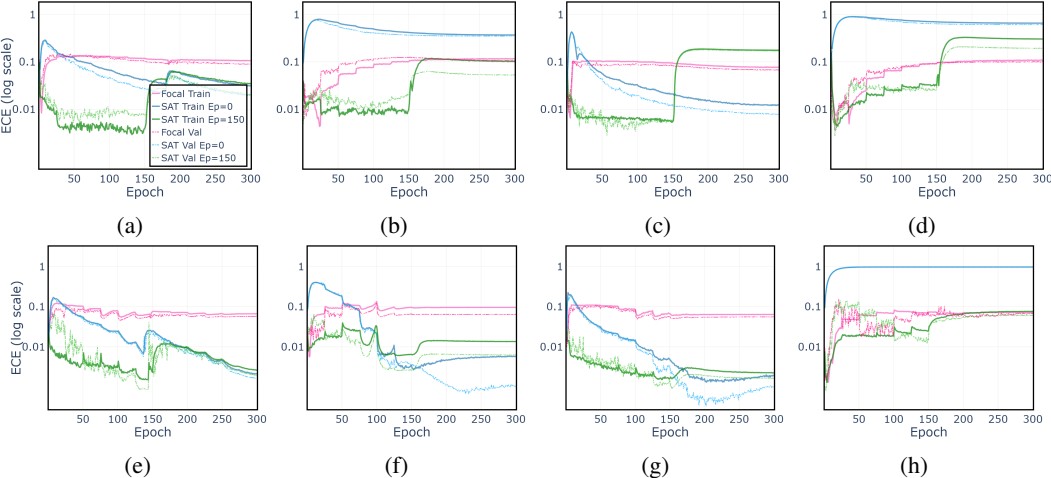

Figure 8: Evolution of the Expected Calibration Error (ECE) across epochs for models trained on CIFAR-10 (a and e), CIFAR-100 (b and f), SVHN (c and g), and Food-101 datasets (d and h) using Focal and Self-Adaptive Training (*first-epochs* and *end-to-end* cases) methods with VGG-16 (a, b, c and d), and ResNet-34 (e, f, g, h) architectures.

**Idk class:** Given that the additional idk class retains the model's knowledge when it does not know, it is reasonable to anticipate that this class will change across epochs, typically exhibiting a decreasing trend. Analyzing the average values of the idk class confidences across epochs provides valuable insights; these plots are shown in Figure 9. Visualising these curves, the idk class appears to

be directly related to calibration. If we compare the ECE across epochs curves with the average of the idk confidences across epochs curves, it is noticeable that both values plot similar trends. When the model believes that it is more certain about what it does not know (indicated by higher average idk confidences), the ECE value tends to be larger, which could be possible due to incorrect confidence values associated with the ground truth classes. This assumption prompts us to consider: *Might the extra idk class approach be beneficial in some way in the calibration aspect or detrimental?* This visualized behaviour was the main source of inspiration to decide to add the predictions associated with the unknown class in the novel Socrates loss to calibrate the training.

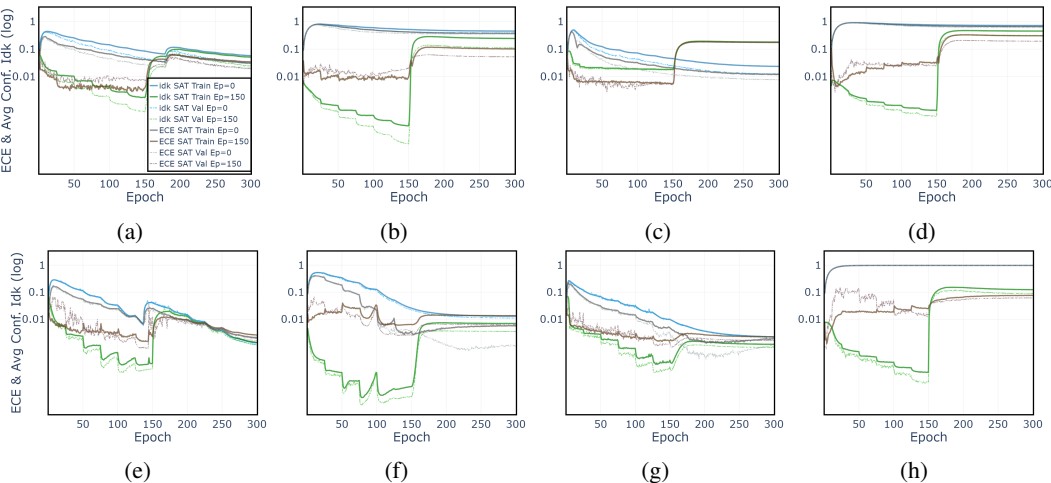

Figure 9: Curves depicting the average values of the idk class confidences across the epochs of models trained on CIFAR-10 (a and e), CIFAR-100 (b and f), SVHN (c and g), and Food-101 (d and h) datasets using Focal and SAT (*first-epochs* and *end-to-end* cases) methods with VGG-16 (a, b, c, and d), and ResNet-34 (e, f, g, and h) architectures.

**Self-Adaptive Training (SAT) loss seems not to be a calibration loss:** Based on the aforementioned empirical analysis the following claim can be made: Unlike Focal loss, which produces very well-calibrated models and follows similar trends across all the datasets and architectures, SAT loss exhibits certain tendencies that ultimately lead to the conclusion that it is not a loss that allows learning calibrated models in all the scenarios, especially when aiming to train for a small number of epochs or when dealing with complex datasets such as Food-101. Additionally, when the loss is used *end-to-end*, the miscalibration in the first epochs is excessively large, and in some cases (CIFAR-100 and Food-101 with VGG-16) it remains significantly large until the end of training. When the loss is applied after the initial epochs (*first-epochs* case), miscalibration begins to emerge.

## G SOCRATES LOSS AS A CALIBRATOR: FIGURES

Due to space constraints, the graphs for all datasets and architectures evaluating the calibration capacity of the Socrates method versus the CCL-SC method are presented in this section. This is supplementary material of Subsection 6.2.

**Accuracy and Loss curves:** The accuracy and loss curves have provided insightful visualizations of performance and fitting. The accuracy curves can be found in Figure 10 and the loss curves in Figure 11.

**ECE values across epochs curves:** The curves plotting the Expected Calibration Error (ECE) values across epochs serve as the focal point of this research, offering key insights into the calibration capacity of the methods. The ECE values across epochs curves can be found in Figure 12.

**Idk class:** Given that the additional idk class retains the model's knowledge when it does not know, it is reasonable to anticipate that this class will change across epochs, typically exhibiting a

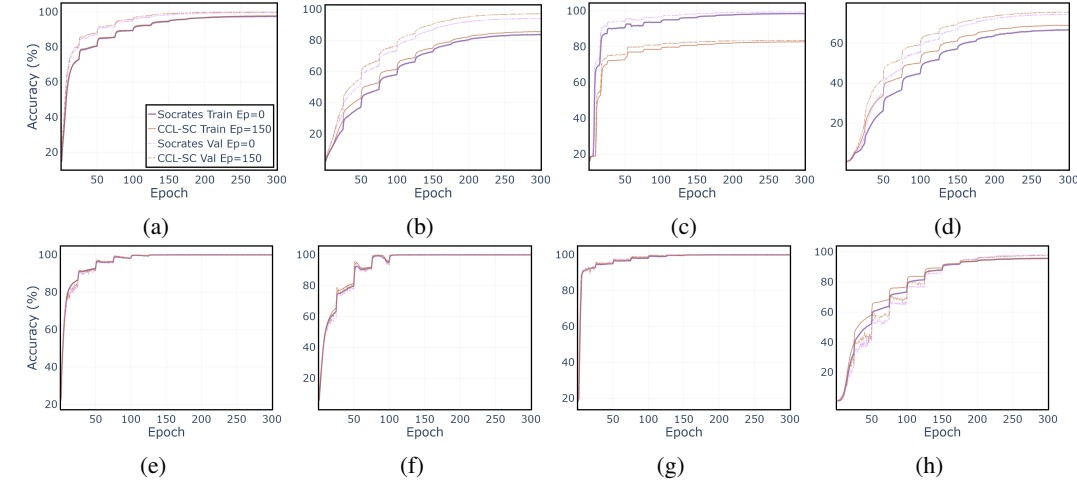

Figure 10: Accuracy across epochs curves of models trained on CIFAR-10 (a and e), CIFAR-100 (b and f), SVHN (c and g), and Food-101 (d and h) datasets using Socrates and CCL-SC methods with VGG-16 (a, b, c and d) and ResNet-34 ( e, f, g, h) architectures.

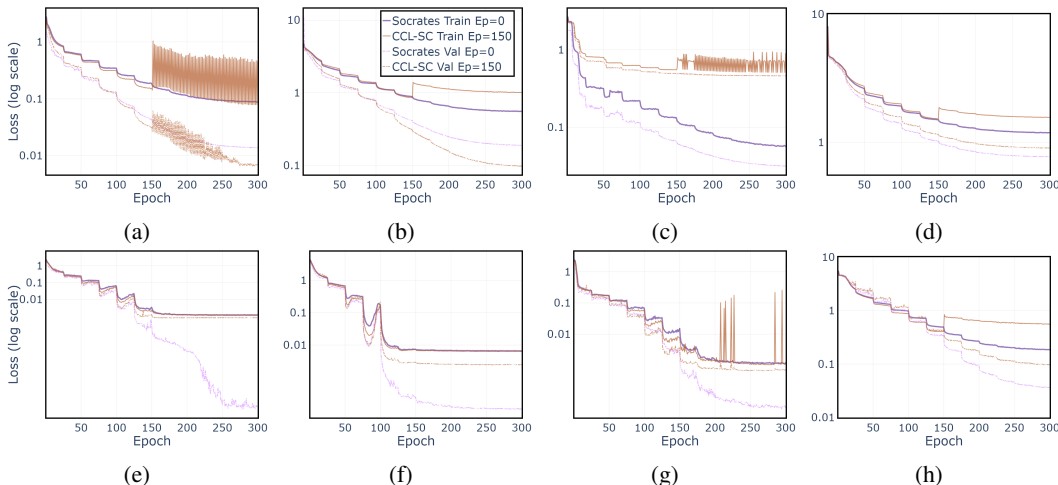

Figure 11: Loss curves of models trained on CIFAR-10 (a and e), CIFAR-100 (b and f), SVHN (c and g), and Food-101 (d and h) datasets using using Socrates and CCL-SC methods (*first-epochs* and *end-to-end* cases) with VGG-16 (a, b, c and d) and ResNet-34 ( e, f, g, h) architectures.

decreasing trend. Analyzing the average values of the *idk* class confidences across epochs provides valuable insights; these plots are shown in Figure 13.

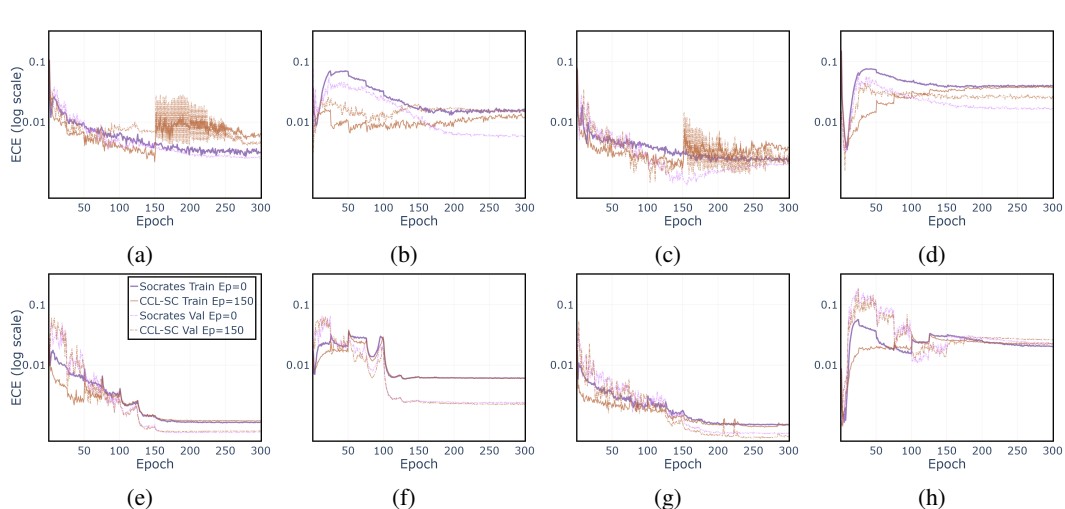

Figure 12: Evolution of the Expected Calibration Error (ECE) across epochs for models trained on CIFAR-10 (a and e), CIFAR-100 (b and f), SVHN (c and g) and Food-101 (d and h) datasets using Socrates and CCL-SC methods (*first-epochs* and *end-to-end* cases) with VGG-16 (a, b, c and d) and ResNet-34 (e, f, g, h) architectures.

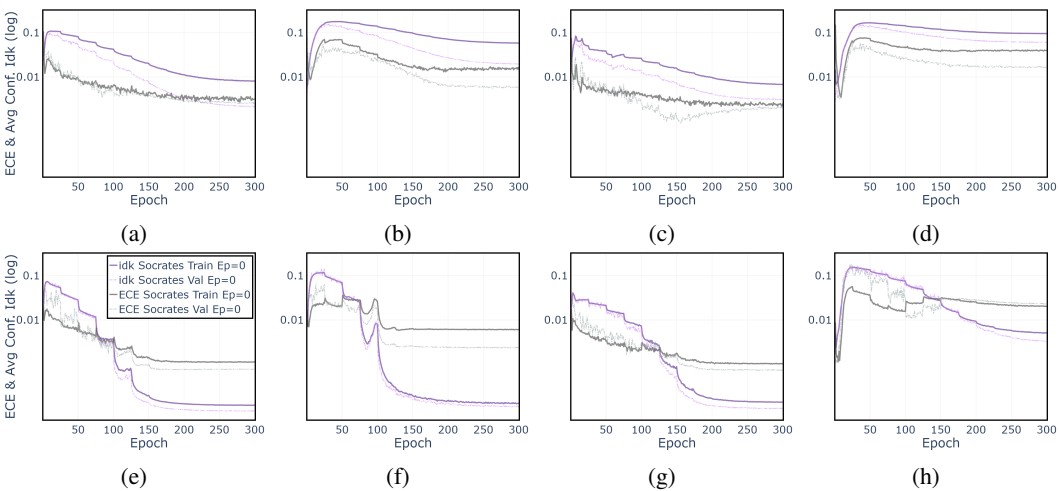

Figure 13: Curves depicting the average values of the idk class confidences across the epochs of models trained on CIFAR-10 (a and e), CIFAR-100 (b and f), SVHN (c and g), and Food-101 (d and h) datasets using Socrates and CCL-SC methods (*first-epochs* and *end-to-end* cases) with VGG-16 (a,b, c and d) and ResNet-34 (e, f, g, h) architectures.

## H    RISK-COVERAGE CURVES

The risk-coverage curves offer a clear representation of the power of Socrates compared to the CCL-SC method. As shown in figure 14, these curves illustrate Socrates reaches similar values or outperforms, thereby providing a more reliable framework for model evaluation.

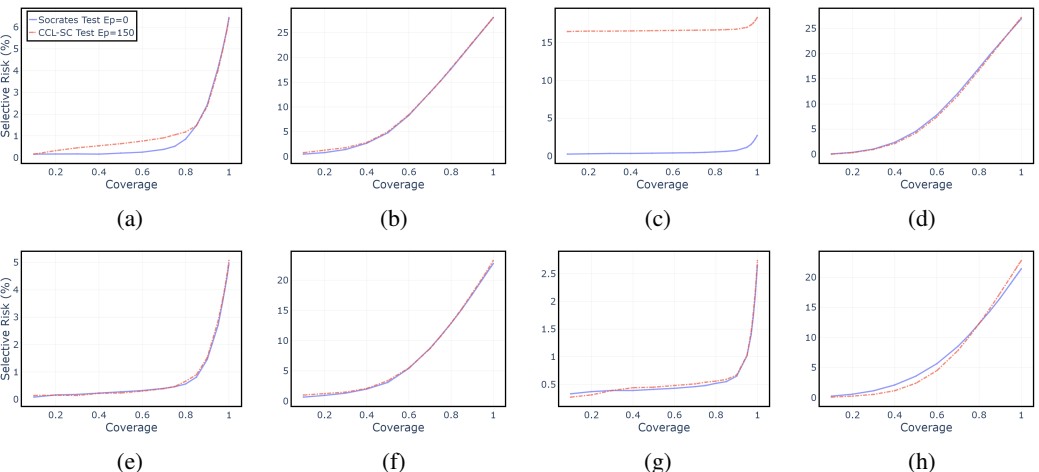

(a)          (b)          (c)          (d)

(e)          (f)          (g)          (h)

Figure 14: Risk-Coverage curves of models trained on CIFAR-10 (a and e), CIFAR-100 (b and f), SVHN (c and g), and Food-101 (d and h) datasets using Socrates (*end-to-end* case) and CCL-SC (*first-epochs* case) methods with VGG-16 (a,b, c and d) and ResNet-34 (e, f, g, h) architectures.

