# OpenReview forum: "Socrates Loss for training ad-hoc calibrated selective classifiers"
_ICLR.cc/2025/Conference — ICLR 2025 Conference Withdrawn Submission_

### Official Review · Reviewer_Nrfp · 2024-10-29

**Soundness:** 2
**Presentation:** 1
**Contribution:** 2
**Rating:** 3
**Confidence:** 3

**Summary:**

The paper proposes Socrates loss which trains both a classifier and a calibrator with a unified optimization goal and presents theoretically analysis on Socrates loss.

**Strengths:**

The combination of confidence calibration and selective classification into a unified training objective is an interesting direction for research. The authors also offer valuable theoretical insights into the Socrates loss, which enhances the understanding of its potential benefits and applications.

**Weaknesses:**

1. The writing quality is poor. For instance:
   - Line 341: "a rise was observed in ECE after the 150 epochs in the first epochs decreasing after convergence."
   - Line 896: "the gamma of the losses was;"
   - Line 868: "E (s) notation"
   - Line 452: "Socrates loss is a suitable loss to output calibrated models."

   I suggest that the authors start each paragraph with a summary sentence, as seen in Section 6.1. Consider using bullet points or numbered lists for observations and claims to improve readbility and emphasis.

2.  The motivation lacks conviction. The abstract claims that *the lack of an unknown class is a drawback of CCL-SC*. However, selective classification can be achieved using baseline methods like SR[1] without needing an unknown class. As noted by Feng et al.[2], *the best-performing selection mechanism should be rooted in the classifier itself, not in the presence of an unknown class*.

3. The paper claims that CCL-SC's is the **only** method combining confidence calibration and selective classification seems overly absolute. Combining the differentiated expected calibration error[3] with selective classification could achieve similar goals.

4. The proposed architecture is limited. Testing other architectures, such as ViT-based methods, and conducting more experiments would add value.  Moreover, as seen in related methods like CCL-SC[4] conducted with ImageNet. Additionally, in resource-limited scenarios, lightweight models could be trained using CLIP model features.

5. In Figure 1, what does "Ep=0" represent? Is it possibly "E_s"?

6. Excessive use of uppercase letters can make the text look cluttered and reduce readability (e.g., Confidence Calibration, Selective Classification, Focal Loss, Weight Decay).

7. An important baseline is missing in Table 2. What about the performance of CE training + SR? Additionally, there's a discrepancy in CCL-SC performance between this paper and the original (both in CIFAR10 and VGG16). For instance, the error rate at 100% coverage in the CCL-SC paper is 5.97±0.11, but it is 6.38 ± 0.14 in this paper. Note that the error rate at 100% coverage is crucial for selective classification error.

8. The rationale for Socrates loss is not convincing. Why exclude the ground truth class probability when computing $\alpha_{dynamic}$? For example, is the performance gain from Socrates loss due to the focal loss term? In later training stages, does $\alpha_{dynamic}$ approach 0, resulting in an approximation to focal loss?

9. What constitutes an acceptable ECE range, as mentioned in Line 340?

10. The primary aim of Socrates loss is to merge calibration and selective classification. However, in Tables 4 and 5, SAT and CCL-SC outperform Socrates loss concerning ECE on most test datasets.

11. The claim that Socrates loss improves calibration seems largely empirical. Is the improvement on hard-to-classify datasets mainly due to the focal loss term?

---

References:

[1] Hendrycks, D. and Gimpel, K. "A baseline for detecting misclassified and out-of-distribution examples in neural networks." In The 5th International Conference on Learning Representations (ICLR), Toulon, France, 2017.

[2] Leo Feng, Mohamed Osama Ahmed, Hossein Hajimirsadeghi, and Amir H. Abdi. "Towards better selective classification." In The Eleventh International Conference on Learning Representations, 2023.

[3] Marx, Charlie, Sofian Zalouk, and Stefano Ermon. "Calibration by distribution matching: trainable kernel calibration metrics." Advances in Neural Information Processing Systems 36, 2024.

[4] Yu-Chang W, Shen-Huan Lyu and Haopu Shang, et al. "Confidence-aware Contrastive Learning for Selective Classification." In Forty-first International Conference on Machine Learning, 2024.

**Questions:**

See the above.

---

### Official Review · Reviewer_gTat · 2024-11-03

**Soundness:** 2
**Presentation:** 2
**Contribution:** 2
**Rating:** 3
**Confidence:** 4

**Summary:**

This manuscript proposed an ad-hoc calibration method to improve the model reliability.

**Strengths:**

The proposed method tries to address an important problem - model reliability.

This manuscript provides detailed theoretical analysis and extensive experiments in multiple datasets.

The proposed method seems to be easy to implement.

**Weaknesses:**

The problem and motivation setting is a bit unclear. The rational behind Socrates Loss needs more clarification.

The results were not sufficiently analyzed. More discussions is needed for each table/figure to provide the justification of the proposed method.

The presentation needs to be improved, for example, the plotted curves in some figures do not have a legend, making it very hard to understand. For example, figure 2 and 3.

Comparison to other methods is limited.

**Questions:**

1. Why do we need to consider an additional unknown class?

2. How's the calibration with unknown class different from transfer learning?

3. What do the coverage rates mean? Why does lower coverage have better performance?

4. Only VGG-16 and ResNet-34 architecture was used as the backbone network, why not use more powerful networks?

5. The loss curve for CCL-SC is vibrating after 150 epochs, which is kind of strange. Is it because CCL-SC is only enabled after 150 epochs? What happens if it is started from 0 epoch, and what happens if we do not use either CCL-SC or Socrates? In Fig 2, it seems brown lines are better than CLL-SC before the 150 epoch.

6. What samples/labels are available in training and testing?

7. How does Ad-hoc methods enhance both accuracy and calibration during training? Does it has access to test samples?

---

### Official Review · Reviewer_BS6y · 2024-11-04

**Soundness:** 1
**Presentation:** 1
**Contribution:** 2
**Rating:** 3
**Confidence:** 4

**Summary:**

This paper introduces a new approach of training ad-hoc calibrated selective classifiers with “Socrates loss”. The proposed method aims to unify classification and confidence calibration into a single optimization framework that includes an unknown class, thereby enhancing model reliability and enabling selective abstention in high-risk predictions. Through experiments on various datasets and architectures, the authors present Socrates Loss as an alternative to existing methods such as SAT and CCL-SC, aiming to improve model calibration throughout training without the need for post-hoc processing.

**Strengths:**

**A new approach:** The paper introduces a new ad-hoc method for training calibrated selective classifiers, which is a considerable approach.

**Weaknesses:**

**Poor readability:** The manuscript is difficult to follow. The main text is not concise and well-organized. Due to the missing explanations (e.g., $t_{i, y_i}$ in eq (1)), the authors' claims are not delivered and justified in the main text.

**Concerns in theoretical analysis:** The theoretical analysis leaves several aspects unclear. In particular, the role of $\Delta_{\text{reg}}$ is unclear. It has a dependence on $\gamma$ and thus the implication of each term in eq (2) is not convincing nor interpretable.

**Concerns in experiments:** There is a lack of clarity around the reported failure of SAT with Food-101. The authors do not sufficiently explore whether this failure might be due to unfair hyperparameter choices or other factors. A clear explanation of the failure of SAT and the success of Socrates could be the main message of this work but is absent in the current manuscript.

**Limited impact:** The performance improvements observed with Socrates Loss are often minor, making the practical advantage of the proposed method unclear. More substantial evidence is needed to demonstrate a meaningful benefit.

**Unclear description of the process of choosing hyperparameters:** The paper does not sufficiently clarify whether hyperparameters were chosen fairly for each baseline, leading to concerns about the fairness of comparisons.

**Questions:**

**$\alpha$'s:** Are $\alpha_{momentum}$ and $\alpha_{dynamic}$ the same?

**Unclear advantage compared to SAT:** Could you summarize the main advantage of the proposed method compared to SAT?

**Figures:** The legends in the figures are too small and the descriptions of axes are incomplete. In addition, Figure 2 can be re-located to the place nearby the part of the main text explaining it.

---

### Official Review · Reviewer_b1Hr · 2024-11-05

**Soundness:** 1
**Presentation:** 1
**Contribution:** 1
**Rating:** 3
**Confidence:** 3

**Summary:**

The paper introduces the Socrates Loss, a novel method aimed at improving model reliability in high-risk applications through ad-hoc calibrated selective classification. Traditional approaches, like Confidence Calibration and Selective Classification, often address reliability issues separately or require post-hoc adjustments.

**Strengths:**

This paper’s contribution lies in integrating both into a single, optimized framework, overcoming limitations found in existing models such as Confidence-aware Contrastive Learning for Selective Classification (CCL-SC) and Self-Adaptive Training (SAT). The Socrates Loss method incorporates an unknown class, allowing the model to abstain from uncertain predictions, and uses a unified loss function that simultaneously calibrates and classifies.

**Weaknesses:**

1.	The paper’s illustrations, specifically Figures 1 and 2, lack clarity in key areas. The labeling in Figure 1 is insufficiently detailed, and there is inconsistency in color schemes, which could hinder comprehension of the data presented. Figure 2 further lacks a legend, which reduces the effectiveness of the visual comparison between methods. Improved and uniform color schemes and more detailed annotations are recommended to ensure the visual clarity of these figures.

2.	The parameters 𝛼_dynamic and 𝛼_momentum appear in the Socrates loss formulation but are not sufficiently distinguished in the text. The similarities in naming make it challenging for readers to discern their distinct roles in the model. Enhanced textual explanations, with a focus on practical and mathematical differentiation between these parameters, would clarify their unique contributions and improve readability.

3.	Key parameters, such as the regularization parameters 𝛾 and 𝛼_dynamic , are briefly introduced without sufficient mathematical justification or detailed explanation of their effects on the model. For instance, while 𝛾 is noted to control the smoothness of the prediction distribution, the specific mathematical mechanisms by which this smooths the output are not elucidated. Similarly, the role of 𝛼_dynamic in balancing the selection function’s weight lacks rigorous theoretical backing. A more thorough derivation of these parameters and their influence on the calibration function would enhance the technical rigor of the discussion.

4.	Although the paper compares the Socrates loss with methods like SAT and CCL-SC, it lacks an analysis of its performance against widely accepted post-hoc calibration techniques such as Platt Scaling and Temperature Scaling. A broader comparative analysis with recent methods below would provide a stronger empirical basis for evaluating Socrates loss’s effectiveness in practical applications.

[1] Meta-Cal: Well-controlled Post-hoc Calibration by Ranking

[2] Mix-n-Match: Ensemble and Compositional Methods for Uncertainty Calibration in Deep Learning

[3] Test Time Augmentation Meets Post-hoc Calibration: Uncertainty Quantification under Real-World Conditions

[4] Pseudo-Calibration: Improving Predictive Uncertainty Estimation in Unsupervised Domain Adaptation

5.	In the theoretical proof presented in the paper, the author employs the Bernoulli inequality to simplify the expression of Socrates loss. While the error terms associated with the Bernoulli inequality are generally negligible in higher-order derivations, their accumulation may have substantial implications when applied to the loss function in deep learning, particularly within high-dimensional spaces and complex data distributions. Such cumulative errors could potentially impact the stability of the training process and the effectiveness of model calibration. However, the paper’s proof lacks a rigorous error analysis, overlooking how error accumulation in high-dimensional spaces may influence model convergence and stability.

6.	Observational results suggest that Socrates loss may not generalize well across all datasets, particularly performing poorly on complex datasets like Food-101. While this issue is noted, it is not fully explored in the text. An in-depth analysis of factors contributing to this performance discrepancy would offer valuable insights for practitioners seeking to apply Socrates loss in diverse real-world applications

**Questions:**

Please refer to my comments in the weakness.

---

### Author Response · Authors · 2024-11-25
**Reply to reviewers**

Dear Reviewers,

Thank you for your reviews. We deeply appreciate your constructive feedback, helpful suggestions, and questions.

The main criticisms indicate a need for better organization, conciseness, and clarity in the paper. We will address these issues in a revised version. We plan to significantly improve Section 4. This revised section will provide a better explanation of the main hyperparameters and their effects. Additionally, we recognise that we assumed certain elements as "known", which we will clarify.

The majority of questions relate to the lack of an ablation and hyperparameter study. We are preparing a series of experiments to address this and will include improved tables and graphs to better illustrate the advantages of our method. Furthermore, we plan to introduce additional comparative methods, such as post-hoc Temperature Scaling and CCL-SC+SR, SAT+SR, Focal+SR, and CE+SR.

We would like to highlight that the main advantage of our method comes from the unified goal (classification and calibration), the use of an unknown class which improves the reliability of the model, and the use of a unique loss implementation. Our Socrates method is notably simple, as it only requires calling and applying the loss function.

As we recognize the need for significant revisions, particularly a comprehensive ablation study, which will require additional time, we have decided to withdraw our submission.

We greatly appreciate your time and consideration. Thank you very much.

---

### Note · Authors · 2024-12-28

I have read and agree with the venue's withdrawal policy on behalf of myself and my co-authors.